# CRISPR-Enabled Autonomous Transposable Element (CREATE) for RNA-based gene editing and delivery

Yuxiao Wang [1,3]✉, Ruei-Zeng Lin[1,3], Meghan Harris [1], Bianca Lavayen[1], Neha Diwanji [1], Bruce McCreedy[1,2], Robert Hofmeister[1] & Daniel Getts [1]✉

## Abstract

**To address a wide range of genetic diseases, genome editing tools that can achieve targeted delivery of large genes without causing double-strand breaks (DSBs) or requiring DNA templates are necessary. Here, we introduce CRISPR-Enabled Autonomous Transposable Element (CREATE), a genome editing system that combines the programmability and precision of CRISPR/Cas9 with the RNA-mediated gene insertion capabilities of the human LINE-1 (L1) element. CREATE employs a modified L1 mRNA to carry a payload gene, and a Cas9 nickase to facilitate targeted editing by L1-mediated reverse transcription and integration without relying on DSBs or DNA templates. Using this system, we demonstrate programmable insertion of a 1.1 kb gene expression cassette into specific genomic loci of human cell lines and primary T cells. Mechanistic studies reveal that CREATE editing is highly specific with no observed off-target events. Together, these findings establish CREATE as a programmable, RNA-based gene delivery technology with broad therapeutic potential.**

**Keywords** Retrotransposon; Gene Editing; Gene Therapy; CRISPR/Cas9
**Subject Categories** Genetics, Gene Therapy & Genetic Disease; Methods & Resources

## Introduction

The recent FDA approval of exagamglogene autotemcel (Casgevy®) for sickle cell disease has marked a significant milestone in CRISPR/Cas9-based gene therapies (Frangoul et al, 2023). Traditional CRISPR/Cas9 systems are commonly used for gene disruption through creating double-strand breaks (DSBs) which are imperfectly repaired via the non-homologous end joining pathway (NHEJ) (Wang and Doudna, 2023). To avoid DSB-associated chromosomal deletions or translocations (Leibowitz et al, 2021; Brunet and Jasin, 2018; Song et al, 2020; Kosicki et al, 2018), Base Editors (BEs) and Prime Editors (PEs) were developed to perform edits by nicking a single DNA strand rather than creating DSBs (Gaudelli et al, 2017; Anzalone et al, 2019). BEs enable targeted single nucleotide conversions, while PEs can insert or replace up to 40 base pairs (bps) without a donor DNA template. Despite these improvements, these editors are limited to perform small edits, which restricts their application in treating most genetic disorders involving a spectrum of mutations, insertions, or deletions across extensive genomic regions.

Expanding on the capabilities of PEs, twin-prime and template-jumping PEs have been developed to integrate larger payloads of a few hundred base pairs (Chen and Liu, 2023). Furthermore, combining PEs with sequence-specific serine integrases has extended this capacity up to 36 kb for the delivery of large DNA fragments encoded in a donor plasmid (Yarnall et al, 2023; Anzalone et al, 2022). Despite their potential, the technical complexity of these systems, which require the co-delivery of multiple components including the prime editor with pegRNA, the serine recombinase, and a DNA donor plasmid, presents substantial challenges for clinical applications (Tang and Sternberg, 2023).

The LINE-1 retrotransposon (referred to as L1) is a mobile genetic element that constitutes 17% of the human genome (Consortium IHGS et al, 2001). It naturally replicates via an RNA intermediate, capable of efficiently reverse-transcribing large mRNA sequences into cDNA which is then integrated into human genome (Beck et al, 2011; Han, 2010). L1 element is transcribed into a bicistronic mRNA encoding two proteins, ORF1p and ORF2p (Beck et al, 2011). ORF1p is an RNA-binding protein that interacts with L1 mRNA transcripts (Khazina et al, 2011). ORF2p is a multi-domain protein containing an endonuclease (EN) and a reverse transcriptase (RT) domain (Mathias et al, 1991; Feng et al, 1996). A distinctive feature of L1 is its 'cis preference,' where ORF1p and ORF2p associate with L1 mRNA transcript to form a ribonuclear protein complex (RNP) (Doucet et al, 2010), facilitating the nuclear import, reverse transcription and insertion of the L1 element (Kojima, 2010; Kulpa and Moran, 2006). The endonuclease domain of ORF2p cleaves at a redundant 5'TTTT/A3' consensus sequence, initiating a target-primed reverse transcription (TPRT) process mediated by the RT domain of ORF2p that converts the L1 mRNA into cDNA (Thawani et al, 2023; Baldwin et al, 2023; Luan et al, 1993). The ORF2p RT domain was shown to be highly processive, capable of reverse transcription of long RNA sequences (Piskareva and Schmatchenko, 2006; Baldwin et al, 2023). However, the consensus motif recognized by the EN domain of L1 ORF2p is widely distributed across the genome, precluding its application for targeted, programmable gene editing. A recent study

[1]Myeloid Therapeutics Inc., Cambridge, MA 02139, USA. [2]Present address: ONK Therapeutics Ltd., Galway, Co. Galway, Ireland. [3]These authors contributed equally: Yuxiao Wang, Ruei-Zeng Lin. ✉E-mail: ywang@myeloidtx.com; daniel@myeloidtx.com

attempting to combine Cas9 with another retroelement R2 was unable to achieve complete retrotransposition and integration (Wilkinson et al, 2023).

Intrigued by the properties of L1, we developed the CRISPR-Enabled Autonomous Transposable Element (CREATE) system co-opting the L1 transposable element for site-specific gene delivery. By combining the precision and programmability of CRISPR/Cas9 with the retrotransposition process of L1, we inserted a 1.1 kb sequence comprising a promoter and green fluorescent protein (GFP) into several sites in the human genome. We further showed that an all-RNA-based delivery approach can achieve effective editing of multiple mammalian cell line and primary human T cells. Together our findings show that the CREATE system is a programmable and highly specific gene delivery platform capable of delivering a payload of over 1 kb with only RNA components, thus offering considerable prospects for diverse therapeutic applications.

## Results

### Design of the CREATE editing system

The CREATE system encodes, on a single mRNA, the codon optimized L1 components ORF1p and a modified ORF2p, separated by the native inter-ORF sequence which mediates an unconventional bicistronic expression of the two proteins (Alisch et al, 2006) (Fig. 1A). To facilitate nuclear import, the ORF2p is fused with an N-terminal SV40 nuclear localization signal (NLS) and a C-terminal nucleoplasmin NLS. Mutation of the catalytic residue within the EN domain of ORF2p (mutation D205A) is a critical modification to prevent cleavage of the degenerate 5'TTTT/A3' consensus motif in the genome by the EN activity of L1 (Feng et al, 1996). The payload cassette consisting of a promoter and the gene of interest is placed at the 3'UTR of the L1 mRNA to form the CREATE mRNA (Fig. 1A). To ensure that the payload cannot be expressed by direct translation prior to genome integration, both the promoter and the payload are encoded in the antisense orientation (Fig. 1A). Critically, the payload is flanked by sequences designed to hybridize with single-guide RNA (sgRNA) target sites in the genome. These sequences are referred to as primer binding site 1 (PBS1) and reverse complement (RC) primer binding site 2 (RC-PBS2), as they serve as binding sites for the genomic DNA 3'-flaps that prime the synthesis of payload cDNA by ORF2p.

Upon cellular uptake, the CREATE mRNA expresses ORF1p and ORF2p proteins, which then co-assemble with the mRNA to form CREATE RNP and enters cell nucleus (Fig. 1A,B). Canonical retrotransposition relies on nicking of the non-target DNA strand by the EN domain of ORF2p (Baldwin et al, 2023; Thawani et al, 2023). To harness this mechanism, a Cas9$^{H840A}$ non-target strand nickase is employed to introduce a single-strand nick guided by sgRNA1 targeting PBS1 in the genome (Fig. 1B, step 1). The liberated DNA 3'-flap then hybridizes with PBS1 on the CREATE mRNA, serving as the primer for the RT domain of ORF2p to initiate first-strand cDNA synthesis (Fig. 1B, step 2). The newly synthesized cDNA strand includes both the payload and the PBS2 in the sense orientation (Fig. 1B, step 3). RNase H2 activity is likely required at this step to remove template mRNA from the RNA:cDNA hybrid (Benitez-Guijarro et al, 2018). Subsequently,

PBS2 on the cDNA hybridizes with the 3'-flap released from the second nick generated by Cas9$^{H840A}$/sgRNA2, and ORF2p utilizes the template jumping mechanism to complete the second strand synthesis (Baldwin et al, 2023) (Fig. 1B, step 4–5). Excision of the original, unedited DNA sequence would result in replacement of the genomic segment between the two sgRNA target sites with the newly synthesized payload cDNA sequence, while avoiding integration of replication competent L1 sequences (Fig. 1B, step 6).

### CREATE-mediated payload gene delivery in mammalian cells

To demonstrate that CREATE can be used to deliver functional genes, we designed a reporter payload cassette consisting of an EF1α core promoter-driven GFP with an SV40 poly(A) signal (totaling 1.1 kb). The AAVS1 site, a well-characterized genomic safe harbor site, was selected to show targeted insertion. We chose two sgRNAs that have been validated previously in the template-jumping PE (TJ-PE) system, targeting sites within AAVS1 that are 90 bp apart (Zheng et al, 2023). A 13 bp PBS1 sequence and 17 bp RC-PBS2 sequence (referred to as "17 bp PBS" in the following sections), also adopted from the TJ-PE study, were positioned on either side of the payload (Fig. 2A).

Both Cas9 and ORF2p are large multi-domain proteins, each exceeding 100 kDa. We were concerned that directly fuse the two proteins would compromise their activities and the ability to enter the nucleus. Since ORF2p could scan genomic DNA for entry sites, we hypothesized that it could independently identify sites nicked by Cas9, eliminating the need for a direct fusion (Tao et al, 2022). To test this concept, we developed a HEK293T cell line engineered to stably express a SpCas9 non-target strand nickase (293T-nCas9$^{H840A}$). Fluorescence-activated cell sorting (FACS) was used to quantify the percentage of cells expressing GFP as a measurement of successful payload integration. Transfection of cells with CREATE mRNA and the two sgRNAs resulted in 0.5% and 0.4% GFP expression on Day 3 and Day 8, respectively (Fig. 2B). No GFP expression was observed in 293T-nCas9$^{H840A}$ cells transfected with non-targeting control sgRNA or in wild-type HEK293T cells (Fig. 2B). These results confirmed sgRNA-guided integration and ruled out the possibility of Cas9-independent, nonspecific insertion. Furthermore, we demonstrated that payload expression required co-transfection of both sgRNAs, as delivery of either sgRNA alone did not result in GFP-positive cells (Fig. 2B). This result indicated that introducing two nicks on opposite DNA strands was necessary for integration. Such requirement inherently enhances specificity, as a single sgRNA off-target binding event would be insufficient to cause payload integration.

FACS was employed to enrich the CREATE edited cells to ~70% GFP positive, after which cells were maintained for an additional two weeks (Fig. 2C). During this time GFP expression remained stable (>70%), indicative of successful and stable genomic integration of the payload. To validate the site-specificity of the insertion and search for potential off-target editing, we performed next-generation sequencing (NGS) with target enrichment by using hybridization probes to capture EF1α-GFP payload sequences from genomic DNA extracted from edited and sorted cells. Insertion sites analysis across the whole genome showed highly specific editing at the intended AAVS1 locus at chromosome 19 (Fig. 2D). Detailed examination of the sequencing reads showed no evidence of off-target integration at other genomic

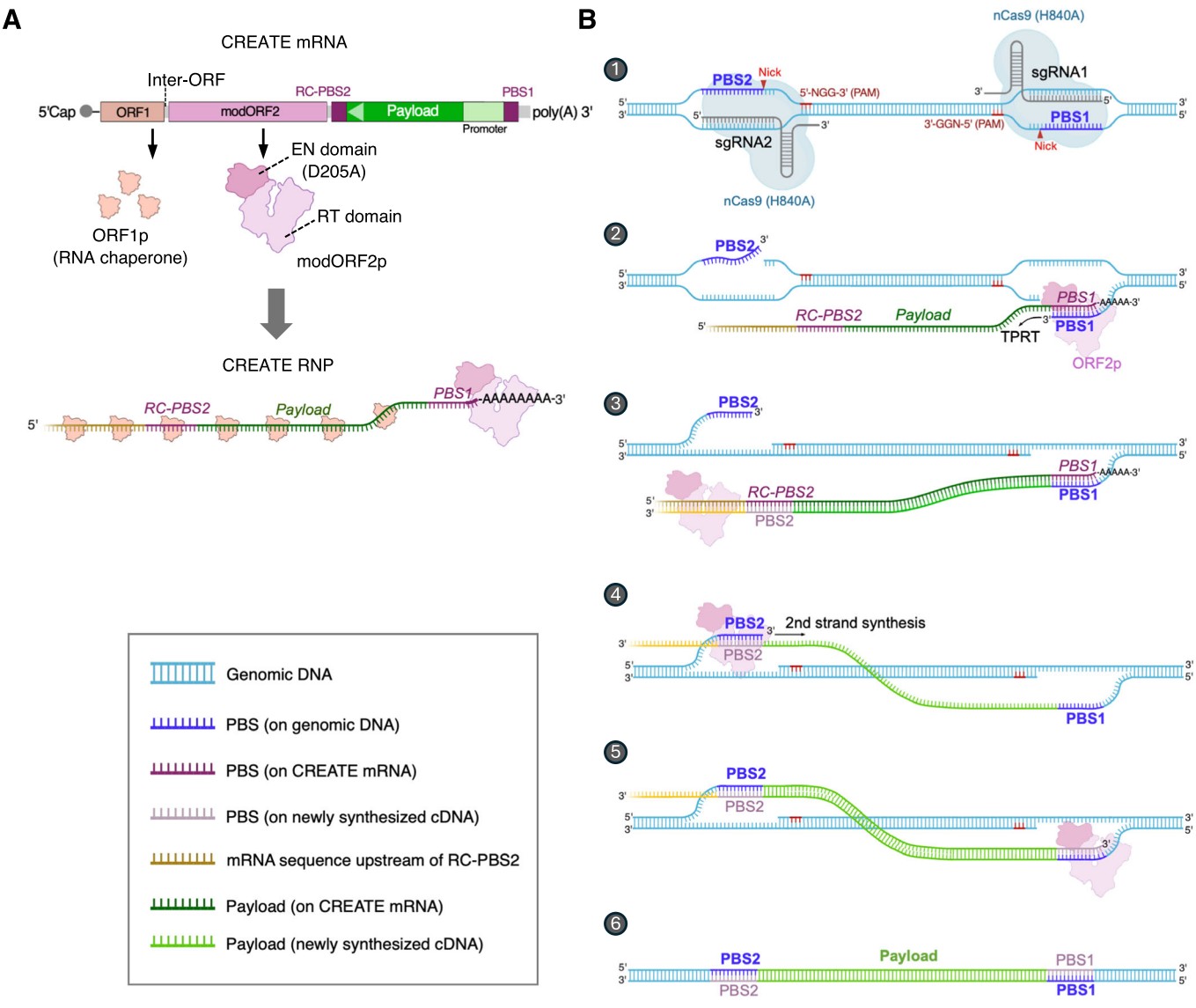

**Figure 1. Design of the CREATE editing system.**

(A) A diagram of CREATE mRNA. ORF2p is modified to inactivate the catalytic residue in the EN domain (D205A) and to fuse with NLS sequences for nuclear import. ORF1p and ORF2p are translated to form CREATE RNP complex. (B) A step-by-step illustration of the mechanism of CREATE-mediated integration of payload gene. (1) Non-target strand nickase Cas9$^{H840A}$ introduces single-strand nicks at two genomic sites directed by sgRNA1 and sgRNA2. (2) The 3'-flap released from the sgRNA1 target site hybridizes with PBS1 on the CREATE mRNA, acting as a primer for the RT domain of ORF2p to initiate first-strand cDNA synthesis. (3) The newly synthesized cDNA strand includes both the payload and the PBS2 in the sense orientation. RNase H2 activity is likely required at this step to remove template mRNA from the RNA:cDNA hybrid. (4) PBS2 on the newly synthesized cDNA anneals with the 3'-flap liberated from the second nick at the sgRNA2 site. ORF2p then employs a template-jumping mechanism to synthesize the second cDNA strand. (5) and (6) Excision of the original, unedited DNA sequence would result in replacement of the genomic segment between the two sgRNA target sites with the newly synthesized payload cDNA.

regions (Fig. 2D; Appendix Table S1). These results highlighted the target specificity of CREATE system.

To further optimize the delivery of CREATE system, sequential transfection of individual RNA components was performed. The specific protocols included (1) Simultaneous transfection of CREATE mRNA and sgRNAs (One-shot Protocol); (2) Transfecting CREATE mRNA four hours prior to sgRNAs (Two-step, mRNA-first Protocol); (3) Transfecting sgRNAs four hours before CREATE mRNA (Two-step, sgRNA-first Protocol). In Protocols (2) and (3), culture media was replaced immediately prior to the second transfection.

Observations supported superior performance of the two-step protocols when compared to the one-shot approach (Fig. EV1A). Notably, the mRNA-first Protocol yielded the highest CREATE editing efficiency, reaching ~1.2% GFP expression on day 3 and remaining stable on day 8. The improvement is likely due to the necessary time for the ORF1p and ORF2p to be translated, co-assemble with CREATE mRNA and subsequently enter the nucleus. Delivery of sgRNAs before the formation of CREATE RNP complex might result in nCas9-induced nicks being repaired before the CREATE editing process can occur. A time interval of 4 to 8 h between mRNA and sgRNA

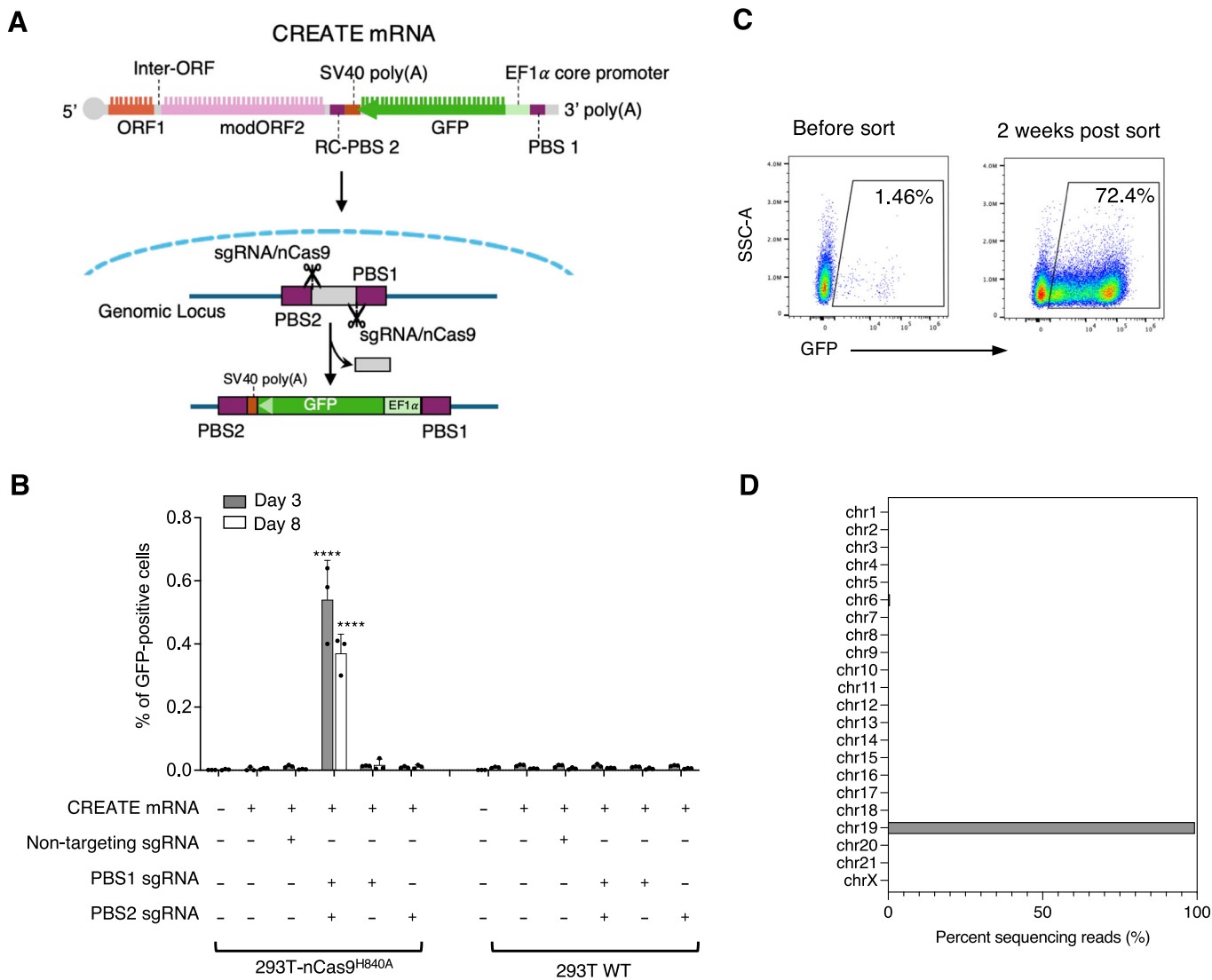

**Figure 2. Site-specific integration of GFP expression cassette in human cells by CREATE.**

(A) A diagram showing CREATE mRNA encoded GFP payload is integrated into AAVS1 locus, replacing a 90 bp segment between the two sgRNA cut sites. (B) CREATE editing requires Cas9 nickase activity and dual sgRNAs. Statistical analysis was performed using two-way ANOVA with Dunnett's multiple comparisons test comparing against the first sample. ****$p < 0.0001$. Statistical significance is labeled for samples with $p < 0.01$. Data are mean ± SD ($n = 3$ biological replicates). (C) Flow cytometry data showing AAVS1 edited cells before and after enrichment for GFP-positive cells. (B, C) experiments were replicated twice. (D) Genome-wide target hybridization sequencing confirms specific integration of GFP payload gene at AAVS1 locus on chromosomal 19. Identified potential sites with >5 sequencing reads were plotted across all chromosomes (also see Appendix Table S1). Exact $p$ values: (B) Day 3 data (from left to right comparing each following sample against the first sample, mock treatment): >0.9999, 0.9997, <0.0001, 0.9992, >0.9999, >0.9999, 0.9972, 0.9992, 0.9990, >0.9999, 0.9987. Day 8 data (from left to right comparing each following sample against the first sample, mock treatment): >0.9999, >0.9999, <0.0001, 0.9977, >0.9999, >0.9999, >0.9999, >0.9999, >0.9999, >0.9999, >0.9999. Source data are available online for this figure.

transfections yielded the highest editing rates of approximately 1.5% (Fig. EV1B). In addition, an increase of sgRNA above 0.242 µg per each well improved the editing efficiency (Fig. EV1C).

## CREATE editing harnesses L1 retrotransposition mechanism

Mutational analysis of the key components of CREATE was performed to investigate the proposed editing mechanisms. We first examined whether CREATE editing requires non-target strand

(Cas9[H840A]) rather than target strand nickase (Cas9[D10A]) activity. When we transfected CREATE mRNA and the two AAVS1-targeting sgRNAs into HEK293T cells stably expressing either Cas9[D10A] or Cas9[H840A] nickase, we observed GFP expression and integration exclusively in the presence of Cas9[H840A] nickase (Fig. 3A), consistent with the model that hybridization of the PBS sequences with 3'-flap released from non-target strand nickase is required for priming cDNA synthesis (Fig. 1B). Of note, utilizing Cas9[H840A] rather than Cas9[D10A] results in the deletion of the PAM sequences for the two sgRNAs after successful insertion, ensuring a

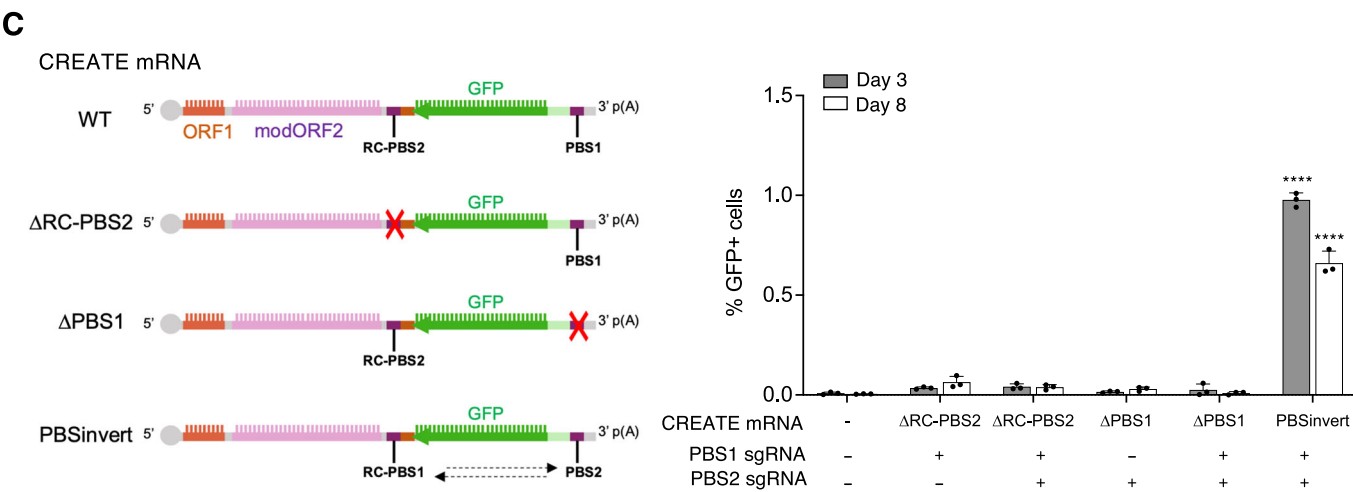

single allele may only be edited once, mitigating concerns about continuous nicking of the genome. Next, we introduced a D702A mutation to inactivate the RT domain of L1 ORF2p (CREATE mRNA^RTdead). This modification entirely abolished the integration and expression of the GFP payload (Fig. 3A), confirming L1 RT activity was required for cDNA synthesis. Similarly, the removal of the entire EN domain (CREATE mRNA^ΔEN) resulted in a marked

decrease in editing efficiency (Fig. 3A), indicating that while the EN domain must be rendered inactive to avert cleavage of the genome at redundant consensus motifs, its complete removal was detrimental to the editing process. This is consistent with the recent reports showing that the EN domain and RT domain of L1 ORF2p are highly integrated structurally and functionally (Thawani et al, 2023; Baldwin et al, 2023).

◄ **Figure 3.  Mechanistic analysis of CREATE editing.**

(A) Mutational analysis of the key enzymatic activities and protein domains involved in editing. Data are mean ± SD ($n = 3$ technical replicates). (B) Influence of the length of PBS sequences in CREATE mRNA and the distance between sgRNA nick sites in the genome on editing efficiency. Data are mean ± SD ($n = 6$ biological replicates). (C) CREATE editing requires two PBS sequences that matches the two sgRNA target sites. Data are mean ± SD ($n = 3$ biological replicates). (A–C) Statistical analysis was performed using two-way ANOVA with Dunnett's multiple comparisons test comparing against the first sample. ****$p < 0.0001$. **$p < 0.01$. Statistical significance is labeled for samples with $p < 0.01$. Experiments were replicated twice. Exact $p$ values: (A) Day 3 data (from left to right comparing each following sample against the first sample, mock treatment): <0.0001, 0.8897, >0.9999, >0.9999. Day 8 data (from left to right comparing each following sample against the first sample, mock treatment): <0.0001, 0.9769, 0.9971, >0.9999. (B) Day 3 data (from left to right comparing each following sample against the first sample, mock treatment): <0.0001, <0.0001, 0.0036, <0.0001, 0.04. Day 8 data (from left to right comparing each following sample against the first sample, mock treatment): <0.0001, <0.0001, 0.0078, <0.0001, 0.0555. (C) Day 3 data (from left to right comparing each following sample against the first sample, mock treatment): 0.5632, 0.3582, 0.9956, 0.8652, <0.0001. Day 8 data (from left to right comparing each following sample against the first sample, mock treatment): 0.0304, 0.3366, 0.6261, 0.9998, <0.0001. Source data are available online for this figure.

## sgRNA target sites and PBS sequences influence editing efficiency

To determine if PBS length impacts insertion efficiency, we designed two additional CREATE mRNA constructs with increasing PBS sequence lengths (30 bp and 50 bp) and compared with the original 17 bp PBS CREATE construct. In these experiments 17–30 bp appeared to support the most efficient editing, while 50 bp was associated with a reduction of GFP integration efficiency (Fig. 3B). It is likely that optimization of PBS length is needed for each specific target locus to improve editing efficiency.

As a search-and-replace tool, CREATE replaces the genomic sequence between sgRNA1 and sgRNA2 with the payload (Fig. 1B). Next, we investigated the impact of the off-set distance between the two sgRNA target sites on editing efficiency. Three sgRNA1s with target sites located 90 bp, 481 bp, or 976 bp from a fixed site targeted by sgRNA2 were designed. While CREATE achieved successful integration of a GFP cassette and replacement of the segment between the sgRNAs in all cases, the editing efficiency dropped gradually with increasing distance between the sgRNAs (Fig. 3B).

## A requirement for paired sgRNAs and matched PBS sequences imparts high fidelity and precision on CREATE editing

To demonstrate the importance of two PBS sites flanking the payload, we designed CREATE mRNA with either PBS1 or a RC-PBS2 removed. In the presence of AAVS1-targeting sgRNAs, the removal of PBS1 or RC-PBS2 resulted in a lack GFP expression on day 3 or day 8 (Fig. 3C). Additionally, exchanging the position of PBS1 and RC-PBS2 while while encoding them in reverse-complementary orientations ("RC-PBS1" and "PBS2) restored the GFP expression (Fig. 3C). The requirement for the two PBS sequences that matched the two sgRNA cut sites was critical for a highly specific integration. When we assessed the top 5 predicted off-target sites for sgRNA1 and sgRNA2 by PCR, no integration of GFP cassette into theses potential sites was observed (Fig. EV2). Together, these data validate the proposed mechanism of CREATE and show that the editing process has several inherent checkpoints that prevent off-target activity. These include: (1) correct base-pairing of PBS1 with sgRNA1 cut site; (2) correct base-pairing of reverse transcribed RC-PBS2 with sgRNA2 cut site; and (3) adjacent sgRNA1 and sgRNA2 nicking.

## CREATE-mediated gene integration into different genomic loci

To determine the ability of the CREATE system to edit and deliver genes specifically into sites in the genome beyond AAVS1, we engineered CREATE to deliver GFP to other genomic loci including HEK3, PRNP, and IDS. The HEK3 locus, situated on chromosome 1 is frequently chosen to benchmark gene editing tools (Zheng et al, 2022; Kong et al, 2021; Kweon et al, 2023). The PRNP locus on chromosome 20 encodes the prion protein implicated in multiple neurodegenerative diseases (Zhang et al, 2016; Palmer et al, 1991), and IDS gene on chromosome X is associated with Hunter's syndrome (Wilson et al, 1990), a lysosomal storage disease. For each targeted site, a pair of sgRNAs 70–90 bp apart were selected. CREATE mRNA, which carried the corresponding 30 bp PBS1 and RC-PBS2 flanking the GFP expression cassette, was deployed to edit each target site. Flow cytometric analysis demonstrated all three sites exhibited comparable editing efficiency, as determined by GFP expression (Fig. 4A).

## Characterization of CREATE editing accuracy by next-generation sequencing

To characterize the editing accuracy at the junction sites between payload sequences and genomic sequence, we performed NGS on the edited and sorted cells by designing PCR primers to amplify the expected junctions followed by Illumina-based short-read sequencing (Fig. 4B). For cells edited with CREATE mRNA with 17 bp PBS targeting AAVS1, we observed ~50% of reads with indels at the PBS1 junction across two independent experiments, while >95% of reads at PBS2 junction were without indels (Fig. 4C). To understand the nature of the indels, we analyzed the five most frequently observed indel variants at the PBS1 and PBS2 junction sites using CRISPResso2 (Clement et al, 2019). At the PBS1 junction, we observed the insertion of a 19 bp sequence that matches the segment between PBS1 and the poly(A) tail on CREATE mRNA (labeled as 3'UTR in Fig. EV3A). This insertion often accompanied with insertion of multiple As and/or duplication of PBS1 sequences (Figs. EV3B and EV4). This suggests that the initiation of reverse transcription can occur via the hybridization of poly(A) tail of CREATE mRNA with the 3'-flap released from single-strand nick generated by nCas9/sgRNA1 (Fig. EV3C). Such model is consistent with previous studies showing that ORF2p RT domain preferentially interact with poly(A) tail of the mRNA, and

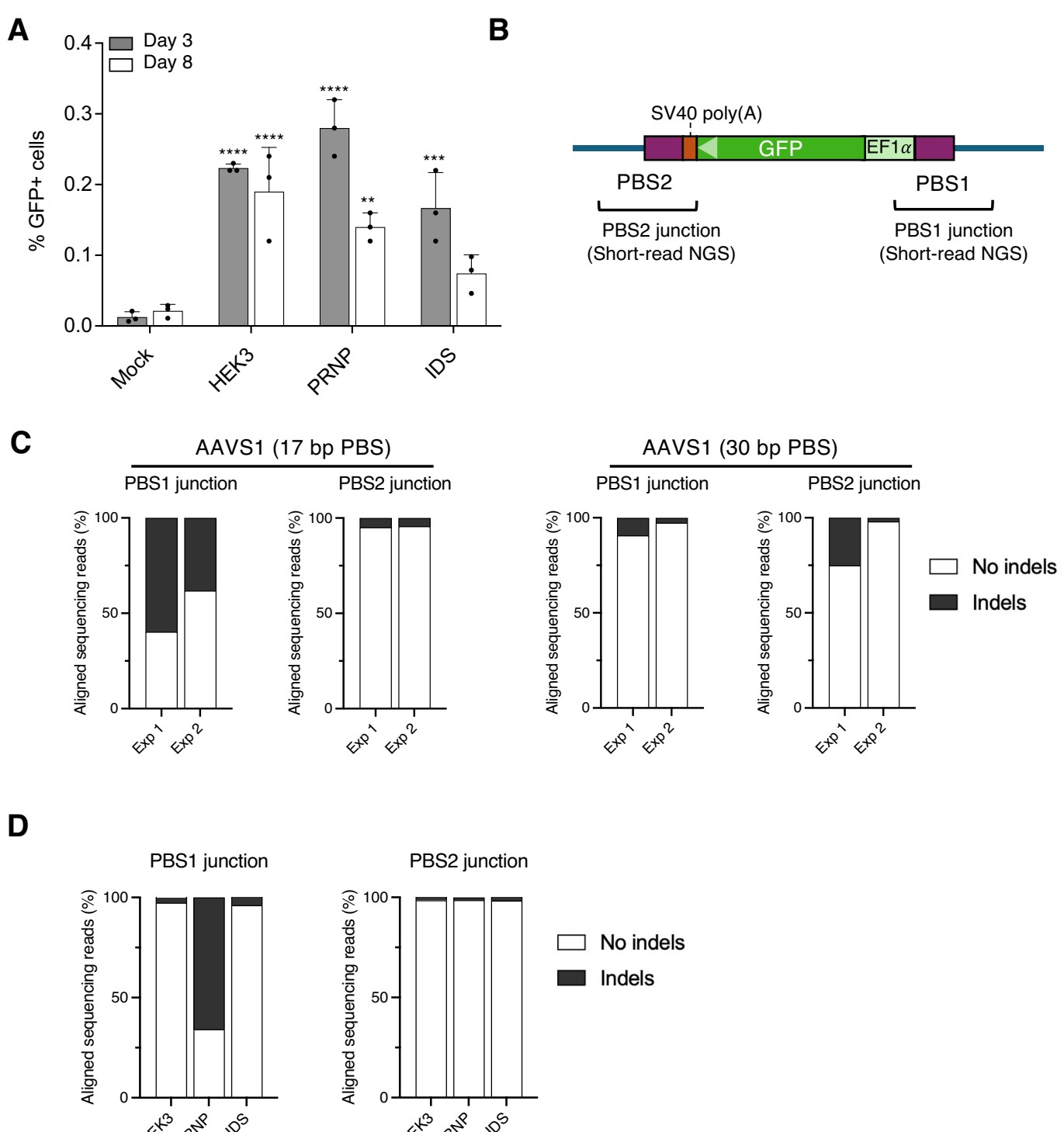

**Figure 4. Integration of GFP into various human genomic loci and NGS analysis of editing outcome.**

(A) Insertion of GFP payload into HEK3, PRNP, and IDS loci by designing unique sgRNAs and matching PBSs. Statistical analysis was performed using two-way ANOVA with Dunnett's multiple comparisons test comparing against the first sample. ****$p < 0.0001$. ***$p < 0.001$. **$p < 0.01$. Statistical significance is labeled for samples with $p < 0.01$. Data are mean ± SD ($n = 3$ biological replicates). Experiments were replicated twice. Exact $p$ values: Day 3 data (from left to right comparing each following sample against the first sample, mock treatment): <0.0001, <0.0001, 0.0001. Day 8 data (from left to right comparing each following sample against the first sample, mock treatment): <0.0001, 0.0016, 0.1770. (B) A diagram showing the amplicon regions subjected to Illumina short-read sequencing. (C) The percentage of indels at the integration junction at AAVS1 locus. Two different CREATE mRNAs were designed with 17 bp or 30 bp PBSs, both targeting AAVS1 locus. Exp 1 and Exp 2 represent two independent experiments. (D) The percentage of indels at integration junctions at HEK3, PRNP, and IDS loci. Edited cells were from experiments shown in (A). Source data are available online for this figure.

can tolerate mismatches between DNA primers and RNA template to initiate TPRT (Monot et al, 2013; Thawani et al, 2023). In addition, we also observed deletions around the PBS1 junction site in the ranger of 40–80 bp (Figs. EV3B and EV4). Notably, the junction at PBS2 was mostly without indels (>95% of the reads) (Figs. 4C and EV4).

Increasing the length of both PBSs to 30 bp greatly reduced indel formation at PBS1, with ~80% of the sequences aligned correctly without indels across two experiments (Fig. 4C; Appendix Fig. S1). In one experiment, PBS2 did not show significant number of indels; while in another experiment we observed deletions of payload sequences adjacent to PBS2 (Appendix Fig. S1D). These results overall suggested that increasing PBS length can suppress formation of indels at PBS junctions.

We also performed the same analysis on the junction regions for cells with targeted insertion of GFP payload at HEK3, PRNP, and IDS sites (Fig. 4A). The results showed that for HEK3 and IDS, the integrations were more than 95% accurate; while the PRNP site showed indel formation only at the PBS1 junction but not PBS2 (Fig. 4D). Analysis using CRISPResso2 showed that the indels at PBS1 are mostly deletions of a poly-T tract from the PRNP genomic sequences, the cause of which is unclear (Appendix Fig. S2C). Overall, analysis of junction site indel formation is consistent with the model that TPRT initiation by the RT domain of ORF2p has a certain degree of error tolerance. However, we also found that PBS2 junction sites do not have such variable indel pattern. In addition, increasing the length of PBSs significantly improved editing accuracy. Similar with PE and related technologies, indels at junction site are likely dependent on the sgRNA choices for the target locus (Anzalone et al, 2022). Optimization of target sgRNAs for each particular target site is necessary to reduce junction indels and improve editing efficiency.

To further confirm that CREATE can achieve full-length transgene integration, we performed PCR using primers designed to hybridize with genomic sequences flanking the two sgRNA target sites (Fig. 5A). Genomic DNA was extracted from CREATE-edited and sorted cells targeting four distinct genomic loci (AAVS1, HEK3, PRNP, and IDS, with 30 bp PBS CREATE constructs) (Fig. 5A). The resulting amplicons were subjected to Nanopore long-read sequencing and aligned to the reference genome to identify reads corresponding to transgene integration between the two sgRNA target sites. Analysis revealed that the majority of reads (>70%) represented full-length or near full-length integrations, defined as those with total indels <20 bp within the integrated payload (Fig. 5A). To further characterize these integrations, we analyzed the distribution of perfect matches, insertions, and deletions across all aligned reads (Fig. 5B–F). Notably, the majority of observed indels were less than 20 bp across all samples (Fig. 5B–F). These results are consistent with the notion that L1 ORF2p RT domain has a high degree of processivity and can mediate integration of large payload (Piskareva and Schmatchenko, 2006).

### CREATE as a fully RNA-based gene delivery tool for different mammalian cell types

Finally, an important attribute of CREATE editing is that all components could be delivered as RNA molecules. To demonstrate this, we synthesized mRNA encoding nCas9$^{H840A}$ with N- and C-terminal NLS. Electroporation was used to deliver the nCas9$^{H840A}$ mRNA, CREATE mRNA and two sgRNAs targeting the AAVS1 locus into HEK293T cell and the immortalized liver cell line Huh7 (Fig. 6A). We found that one-time electroporation led to ~1% and 0.2% GFP payload re-expression in Huh7 and HEK293T cells, respectively (Fig. 6B). To demonstrate that CREATE can be applied to edit primary cells, we isolated human T cells from health donor blood, which were activated using anti-human CD3/CD28 antibodies complexes for 2 days. One-time electroporation of T cells with CREATE mRNA, Cas9$^{H840A}$ mRNA and two sgRNAs resulted in stable integration and re-expression of GFP payload in ~0.3% of the cells (Fig. 6C). In an attempt to reduce the overall size of the CREATE system and improve delivery efficiency, we deleted ORF1p from the CREATE construct. Electroporation of ΔORF1 variant of CREATE mRNA did not result in successful integration in primary T cells, suggesting that exogenously expressed ORF1p is important for efficient integration (Fig. EV5A,B). Taken together, our results highlight the potential to develop CREATE into a fully RNA-based gene delivery tool for in vitro and in vivo applications.

## Discussion

Here we have engineered CREATE, a programmable, site-specific tool co-opting the human L1 retrotransposon for integration of full-length genes into the human genome. Key design elements for the site specificity of CREATE include the modification of the L1 ORF2p endonuclease and the introduction of a Cas9 nickase, which does not directly cause DSBs. Two primer binding sites flanking the payload are critical for the initiation of reverse transcription and converting the payload RNA into cDNA for integration into the genome. To our knowledge, we describe for the first time the use the human L1 to deliver genes with specificity and programmability. An earlier study explored the direct fusion of a Cas9 or Cas12a endonuclease with the reverse transcriptase domain of L1 ORF2p for site-specific integration (Manoj et al, 2021). While Cas-directed cleavage and subsequent RNA payload integration was achieved, the study was limited to payload insertion into high-copy plasmids in *E. coli*. The use of a wild-type Cas9 endonuclease exhibited significant toxicity. In contrast, CREATE employs a Cas9 nickase in combination with the full-length, endonuclease inactivated ORF2p for targeted integration. Recent high-resolution cryo-EM studies that investigated ORF2p complexed with DNA and RNA substrates revealed mechanisms of target site recognition and initiation of reverse transcription (Baldwin et al, 2023; Thawani et al, 2023). Multiple domains in ORF2p fold into an integrated structure that wrap around the substrates to effectively couple genomic DNA binding, target site cleavage and TPRT. Our findings support the notion that preserving all essential domains of ORF2p can facilitate productive integration in mammalian genomes.

L1 has been reported to be capable of endonuclease-independent (ENi) retrotransposition, during which L1 element inserts into DNA lesions that remain unrepaired with no apparent target site sequence preference (Morrish et al, 2007, 2002). In addition, a recent study showed that L1 can spontaneously invade Cas9 created DSBs and, at a very low frequency also into Cas9 nickase induced single strand breaks (Tao et al, 2022). In our study, NGS analysis of the insertion junctions from edited cells showed that when a short (13 bp) PBS1 was used, indels were frequently observed at PBS1

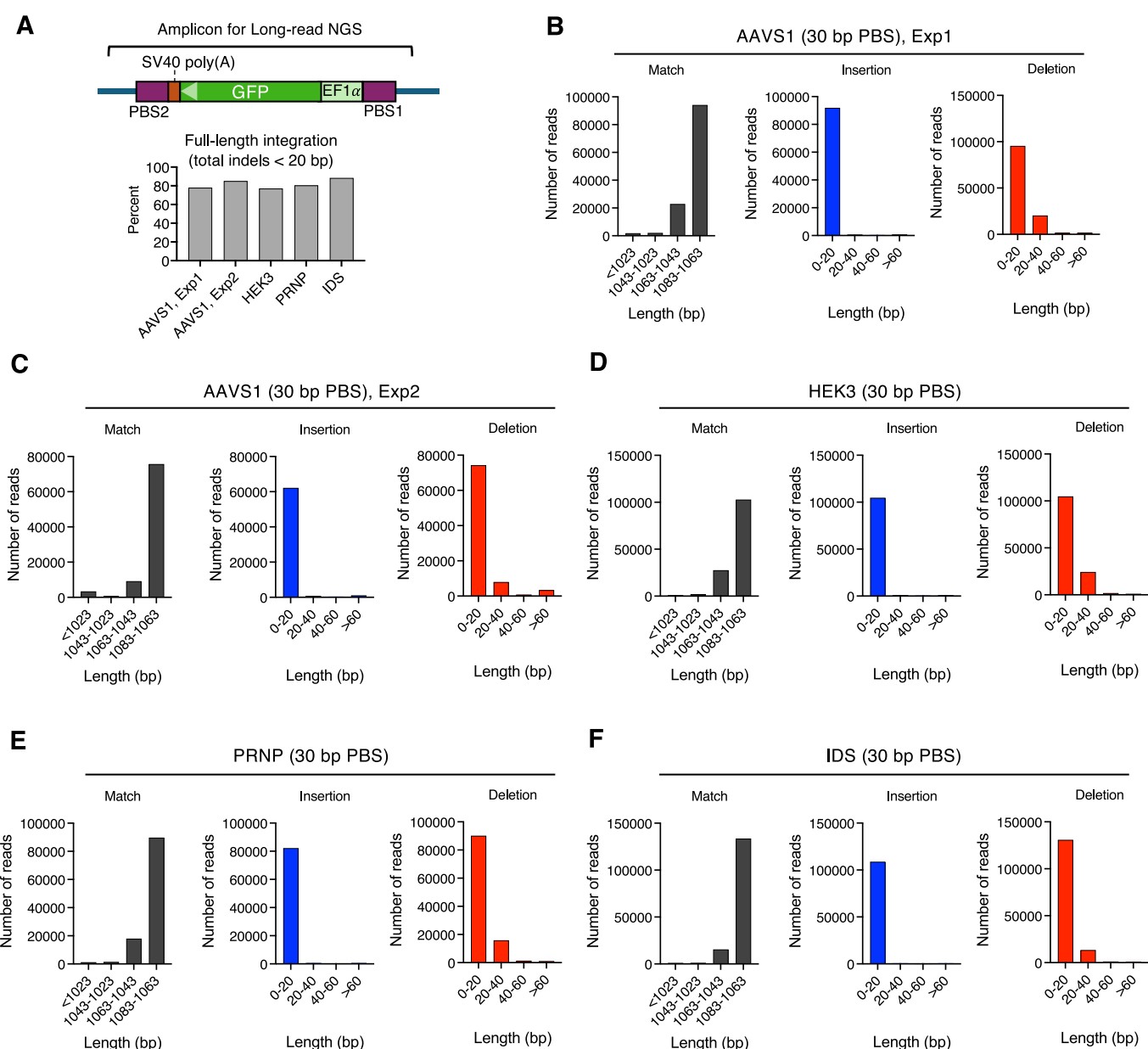

**Figure 5. Long-read NGS analysis of payload transgene integration.**

(A) Analysis of the percentages of full-length and near full-length integrations across four genomic loci. (B–F) Length distribution of perfect matches, insertions, and deletions in aligned reads from long-read NGS analysis. (B, C) Analysis results for cells edited with CREATE mRNA (30 bp PBS) targeting AAVS1 from two independent experiments. (D) Analysis results for cells edited with CREATE mRNA (30 bp PBS) targeting HEK3. (E) Analysis results for cells edited with CREATE mRNA (30 bp PBS) targeting PRNP. (F) Analysis results for cells edited with CREATE mRNA (30 bp PBS) targeting IDS. Source data are available online for this figure.

junction that resembles the outcome of L1 integration via ENi pathways. However, when PBS length was increased to 30 bp, the fidelity of the editing was much improved, indicating that the longer PBS length promoted specific hybridization with the 3′DNA flap released from nCas9 nicks for initiation of TPRT. In addition, it is worth noting that ENi pathway of L1 was observed to occur mostly in cells that lack NHEJ DNA repair mechanism or has dysfunctional telomere (Morrish et al, 2007, 2002). We showed that CREATE can integrate into primary T cells and 293T cells, both of which do not have such deficiencies. Moreover, single-strand nicks

created by the D10A variant of Cas9 nickase did not result in integration, thus excluding non-specific invasion of DNA lesions as the dominant mechanism of integration. We propose that while non-specific invasion of DNA lesions may be an intrinsic feature of L1 ORF2p, optimization of sgRNA target choices and PBS sequences can suppress such outcome and improve editing accuracy.

Non-LTR retroelements have gained increasing interest for the delivery of larger payloads. R2 retroelements, found in metazoans and birds, have been used to insert large RNA payloads (up to 4 kb)

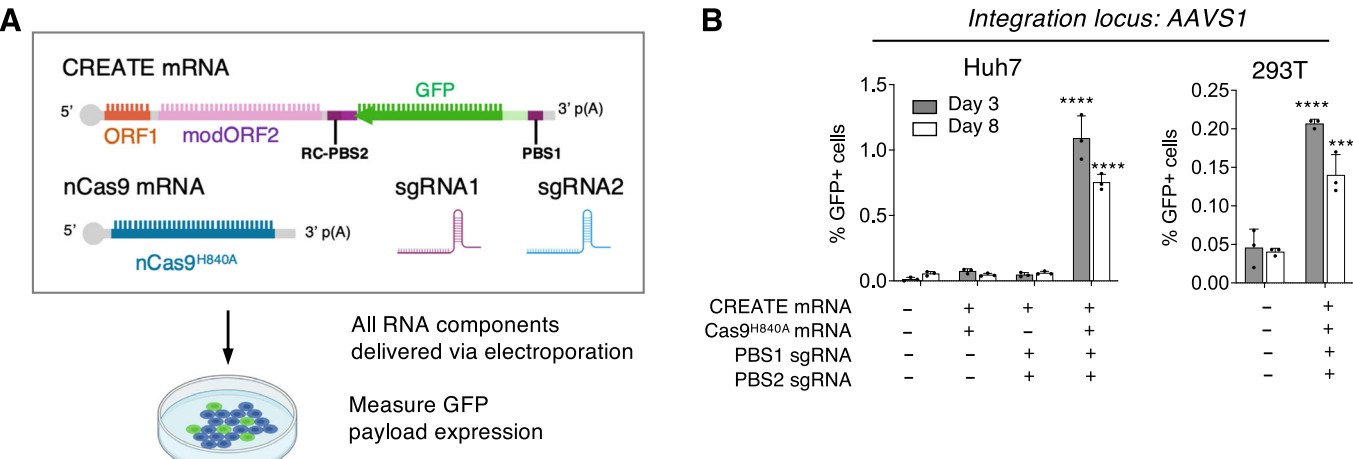

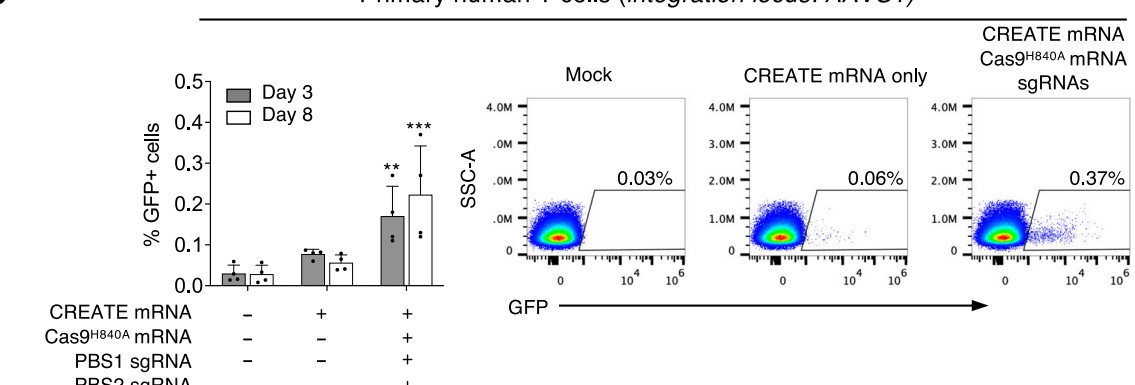

**Figure 6. all RNA-based delivery of CREATE for editing of mammalian cell lines and primary human T cells.**

(A) Co-delivery of CREATE editing components as mRNA/sgRNAs. (B) RNA-based GFP payload insertion at AAVS1 locus in Huh7 and HEK293T cells. Statistical analysis for Huh7 cell result was performed using two-way ANOVA with Dunnett's multiple comparisons test comparing against the first sample. Statistical analysis for HEK293T cell result was performed using two-way ANOVA with Šídák's multiple comparisons test comparing between the two conditions. Data are mean ± SD ($n = 3$ biological replicates). Experiments were replicated three times. Exact $p$ values: Huh7 cells Day 3 data (from left to right comparing each following sample against the first sample, mock treatment): 0.5337, 0.8605, <0.0001. Day 8 data (from left to right comparing each following sample against the first sample, mock treatment): 0.9957, 0.9994, <0.0001. 293T cells Day 3: <0.0001. Day 8: 0.0003. (C) One-step electroporation with all RNA components results in stable integration of a GFP expression cassette in primary T cells. Statistical analysis was performed using two-way ANOVA with Dunnett's multiple comparisons test comparing against the first sample. **$p < 0.01$. ***$p < 0.001$. Statistical significance is labeled for samples with $p < 0.01$. Data are mean ± SD ($n = 4$ biological replicates). Experiments were replicated three times. Exact $p$ values: Day 3: 0.4330 (Mock vs CREATE mRNA only), 0.0068 (Mock vs all components). Day 8: 0.7333 (Mock vs CREATE mRNA only), 0.0004 (Mock vs all components). Source data are available online for this figure.

into eukaryotic genomes (Chen et al, 2024; Zhang et al, 2024). However, R2 elements are naturally restricted to the ribosomal DNA (rDNA) loci, which limits their utility for editing in other genomic locations. In addition, insertion at rDNA site may result in reduced expression over time due to rDNA array instability (Zhang et al, 2024). An attempt to integrate R2 elements with Cas9 for RNA-guided reverse transcription and integration did not achieve complete cDNA synthesis and integration (Wilkinson et al, 2023). Exploiting the replication mechanisms of L1 element, CREATE achieved an integration efficiency of ~1.5% in multiple mammalian cell lines, inserting a 1.1 kb GFP expression cassette without detectable off-target integration in multiple genomic sites.

Importantly, we also demonstrated the integration of the GFP cassette into primary human T cells, highlighting the promise of L1 as a tool to engineer immune cells. Given L1's natural capacity to mediate insertion of ~6 kb of L1 RNA, we anticipate that future iterations of CREATE can deliver even larger sequences.

The L1 element has been associated with increased risk of cancer and cellular senescence (Xiao-Jie et al, 2016; Gorbunova et al, 2021). A gradual loss of epigenetic suppression of L1 transcription can lead to reactivation of L1 retrotransposition which may cause insertional mutagenesis or chronic inflammation triggered by a type I interferon response (Cecco et al, 2019; Mendez-Dorantes and Burns, 2023; Grundy et al, 2022). In the present study, no off-target

integrations were observed suggesting that CREATE does not cause genomic instability. Several properties and modification ensure that the CREATE system integrates the payload gene without propagating L1 components or perturbing their repressed state. CREATE mRNA is only transiently expressed and once on-target integration has occurred, the sgRNA PAM sequences are deleted which prevents further re-editing. Finally, the EN domain of ORF2p in the CREATE system is inactivated to prevent non-specific cleavage of genomic DNA.

Both CREATE and PE utilize reverse transcription-based mechanism to convert RNA-encoded edits to cDNA which is then incorporated into the genome. PE and related technologies, when used without supplying a DNA donor plasmid, are limited to small edits of less than 100 bp (Chen and Liu, 2023). For example, Anzalone et al demonstrated that the PE3 system could achieve insertion of short, 12 bp and 36 bp sequences at moderate efficiencies (32% and 17%, respectively); however, the efficiency dropped to 0.8% for a 108 bp insertion, and longer sequences were not attempted (Anzalone et al, 2022). To overcome such limit, current state-of-the-art approaches such as PASTE invariably require the co-delivery of a DNA recombinase and a donor plasmid to achieve gene-sized payload insertion (Yarnall et al, 2023). In contrast, the CREATE system represents the first fully RNA-based approach for gene delivery, capable of inserting a 1.1 kb gene cassette into multiple genomic loci without relying on DNA templates. This eliminates the need for viral vectors to deliver DNA donor plasmids, which pose safety risks and manufacturing challenges. We anticipate that screening L1 variants with improved retrotransposition activity can further increase the editing efficiency. Furthermore, the combination of the modified L1 element with alternative programmable RNA-guided DNA endonucleases such as Cas12, TnpB and Fanzor can be explored to expand the capability of the CREATE system (Zetsche et al, 2015; Saito et al, 2023; Karvelis et al, 2021).

Varying degrees of indel formation at the transgene/genomic locus junction were noted for CREATE-edited cells. This phenomenon has also been described for PE-related technologies that utilize a pair of sgRNAs in close proximity, such as in the PE3 system and GRAND editing. 10–20% indel by-products were observed for the PE3 system (Anzalone et al, 2019). Using GRAND editing to insert an 150 bp fragment to various genomic loci yielded 5.8% to 63.0% accurate editing with substantial indel formation (Wang et al, 2022). In addition, single-strand nicks adjacent to each other can result in DSB formation, leading to NHEJ pathway mediated indels (Ran et al, 2013). It has been shown that if the target sites of two nicking sgRNAs have an off-set distance of >150 bp, DSB formation can be avoided (Ran et al, 2013). Systemic testing of sgRNAs and their off-set distance can identify the optimal design that maximize editing efficiency while minimizing the potential formation of indels and DSBs.

In summary, the development of CREATE combines the gene integration capacity of a human L1 retroelement with the CRISPR/Cas9 precision. As a fully RNA-based genome engineering tool that does not require DNA templates, it complements existing approaches and has the potential to address a large variety of genetic diseases.

# Methods

### Reagents and tools table

| Reagent/Resource | Reference or Source | Identifier or Catalog Number |
|---|---|---|
| **Experimental models** | | |
| HEK293T cell line | ATCC | CRL-3216 |
| Huh7 cell line | JBRC | JCRB0403 |
| Human Primary T cells from Donor Leukopak | Generated at Myeloid Therapeutics | N/A |
| **Recombinant DNA** | | |
| Plasmids for CREATE mRNA synthesis | This study | Appendix Table S2 |
| **Oligonucleotides and other sequence-based reagents** | | |
| AAVS1 PBS1 sgRNA | GAUGGAGCCAGAGAGGAUCC | |
| AAVS1 PBS2 sgRNA | GCAGCUCAGGUUCUGGGAGA | |
| AAVS1 PBS1 (481-bp replacement) sgRNA | GCUCUUCCAGCCCCCUGUCA | |
| AAVS1 PBS1 (976-bp replacement) sgRNA | CCUUCCCUGCCGCCUCCUUC | |
| HEK3 PBS1 sgRNA | GUCAACCAGUAUCCCGGUGC | |
| HEK3 PBS2 sgRNA | GGCCCAGACUGAGCACGUGA | |
| PRNP PBS1 sgRNA | GCAGUGGUGGGGGGGCCUUGG | |
| PRNP PBS2 sgRNA | GCAUGUUUUCACGAUAGUAA | |
| IDS PBS1 sgRNA | GCAUUUUCGAUUCCGUGACU | |
| IDS PBS2 sgRNA | ACUGAGGGAUGUCUGAAGGC | |
| Non-targeting sgRNA | AAAUGUGAGAUCAGAGUAAU | |
| PCR primers for amplicon NGS analysis | Genewiz | Appendix Table S3 |
| PCR primers for off-target detection | Genewiz | Appendix Table S4 |
| **Chemicals, Enzymes and other reagents** | | |
| Dulbecco's modified Eagle's medium (DMEM) | ThermoFisher | 10566016 |
| Penicillin/Streptomycin/ Glutamine | ThermoFisher | 10378016 |
| Fetal Bovine Serum | ThermoFisher | A5670701 |
| TrypLE Select | ThermoFisher | 12563011 |
| EasySep Human T cell isolation kit | StemCell Technologies | 17951 |
| ImmunoCult™ Human CD3/CD28 T Cell Activator | StemCell Technologies | 10991 |
| T Cell TransAct™, human CD3/CD28 T cell Activator | Miltenyi Biotec | 130-128-758 |
| TexMACS | Miltenyi Biotec | 170-076-309 |
| Heat-inactivated human AB serum | Geminibio | 100-512 |

| Reagent/Resource | Reference or Source | Identifier or Catalog Number |
|---|---|---|
| EDTA | ThermoFisher | AM9260G |
| IL2 | ThermoFisher | 200-02-100UG |
| IL7 | ThermoFisher | 200-07-50UG |
| IL15 | ThermoFisher | 200-15-50UG |
| Hygromycin B | ThermoFisher | 10687010 |
| Monarch Genomic DNA Purification Kit | New England BioLabs | T3010L |
| Qiagen DNeasy Blood & Tissue Kits | Qiagen | 69504 |
| Monarch® DNA Gel Extraction Kit | New England BioLabs | T1120L |
| Q5 High-Fidelity 2X PCR Master Mix | New England BioLabs | M0494L |
| Q5 site-directed mutagenesis kit | New England BioLabs | E0554S |
| NEBuilder® HiFi DNA Assembly Master Mix bly mix | New England BioLabs | E2621L |
| QIAprep Spin Miniprep Kit | Qiagen | 27106 |
| HpaI | New England BioLabs | R0105L |
| HiScribe T7 mRNA Kit with CleanCap Reagent AG | New England BioLabs | E2080S |
| GeneJET PCR Purification Kit | Thermo Fisher Scientific | K0702 |
| Monarch® RNA Cleanup Kit | New England BioLabs | T2050L |
| E. coli poly(A) polymerase | New England BioLabs | M0276L |
| TapeStation RNA ScreenTape Analysis | Agilent | 5067-5576 |
| Lipofectamine MessengerMAX | ThermoFisher | LMRNA015 |
| Lipofectamine RNAiMax | ThermoFisher | 13778075 |
| MaxCyte OC25x3 cuvette | MaxCyte | SOC25x3 |
| OptiMEM | ThermoFisher | 31985062 |
| **Software** | | |
| FlowJo v10 | https://www.flowjo.com/ | |
| GraphPad Prism v10 | https://www.graphpad.com/ | |
| CRISPResso2 | https://github.com/pinellolab/CRISPResso2 | |
| **Other** | | |
| Cytek Northern Lights Flow Cytometer | Cytek | |
| BD FACSMelod Cell Sorter | BD Biosciences | |
| Agilent TapeStation | Agilent | |
| MaxCyte ATx Electroporator | MaxCyte | |

## Cell culture

HEK293T (ATCC, cat. no. CRL-3216) and Huh7 cells (JBRC, Cat. No. JCRB0403) were maintained in Dulbecco's modified Eagle's medium (DMEM) supplemented with 10% (v/v) fetal bovine serum (Thermo-Fisher) and 1% (v/v) Penicillin/Streptomycin/Glutamine (Thermo-Fisher). Cells were cultured at 37 °C with 5% $CO_2$. TrypLE Select (ThermoFisher) was used for passaging and harvesting cells. Human primary T cells were isolated from donor leukopak by negative immunomagnetic selection using EasySep™ Human T Cell Isolation Kit (STEMCELL Technologies). T cells were activated using ImmunoCult™ Human CD3/CD28 T Cell Activator (STEMCELL Technologies) for two days in TexMACS media with 5% heat-inactivated human AB serum, 10 ng/mL IL-2, 5 ng/mL IL-7, and 5 ng/mL IL-15 (all from ThermoFisher) prior to electroporation.

## sgRNAs

sgRNAs were acquired from Synthego or Thermo Fisher Scientific and reconstituted with TE buffer into 100 pmol/μL stock solution. Sequences of sgRNAs are listed in Reagents and Tools table.

## Generation of Cas9-expressing HEK293T cell lines

PiggyBac vectors expressing Cas9 nickase H840A or D10A variants with an N-terminal SV40 NLS and a C-terminal Nucleoplasmin NLS under the CMV promoter were constructed by Vector Builder. PiggyBac vectors (1.25 μg) and hyPBase mRNA (0.7 μg) were co-transfected into $2.5 \times 10^6$ HEK293T cells using MaxCyte ATx electroporator by following the manufacturer's protocol. Three days after transfection, cells were selected with hygromycin (400 μg/mL) for 10 days, to acquire stable Cas9-expressing HEK293T cell lines.

## Cloning of CREATE constructs

DNA sequences encoding human L1 ORF1p (GenBank: AAB59367.1) and ORF2p (GenBank: AAA51622.1) with N-terminal SV40 NLS and C-terminal nucleoplasmin NLS were codon optimized and synthesized at Twist Bio. Endogenous L1 interORF region sequences flanked by restriction sites PshAI (GACAGCCGTC) and AccI (GTATAC) was synthesized at Twist Bio. The full constructs containing all L1 components including ORF1p, interORF, and ORF2p were then assembled using Hifi Assembly mix (NEB) into pCMV vector with the CMV promoter region removed. A T7 promoter was introduced upstream of ORF1p for in vitro transcription. The payload cassette including an EF1α core promoter, EGFP and SV40 poly(A) signal was synthesized by Integrated DNA Technologies and inserted at the 3'UTR of the L1 mRNA using Hifi Assembly mix (NEB). gBlocks gene fragments containing different PBS site sequences were synthesized by Integrated DNA Technologies, and subsequently cloned into the vector to flank the payload gene. Sequences of PBS sites and payloads are listed in Appendix Table S2. Mutation of ORF2 RT domain was introduced using Q5 site-directed mutagenesis kit (NEB). Deletion of ORF2 EN domain was performed by PCR and ligation cloning.

## mRNA synthesis via in vitro transcription

Plasmid templates for in vitro transcription were linearized by restriction enzyme and templates were purified by column clean up (GeneJET PCR Purification Kit, Thermofisher) and eluted in 40 µl of nuclease-free water. All in house generated mRNA was transcribed via T7 polymerase and co-transcriptionally capped utilizing New England Biolabs HiScribe T7 mRNA Kit with CleanCap Reagent AG according to the manufacturer's protocols. The in vitro transcription reaction was allowed to proceed for 2 h at 37 °C. Reaction mixtures were treated with DNase I and incubated at 37 °C for 30 min before immediate purification (Monarch® RNA Cleanup Kit, NEB). A sample of each mRNA was reserved for quality analysis and transcripts were tailed with *E. coli* poly(A) polymerase (NEB) at 37 °C for 30 min, and subsequently purified (Monarch® RNA Cleanup Kit, NEB). All tailed and untailed mRNA samples were normalized to a concentration of 50 ng/µl and quality was assessed via TapeStation RNA ScreenTape Analysis (Agilent, RNA ScreenTape, 5067-5576) according to the manufacturer's protocol.

## Lipofectamine transfection

HEK293T cells were seeded on 24-well plates at 80,000 cells the day prior to transfection in DMEM with 2% FBS. Transfection was performed by either one of these protocol:

- One-shot Protocol (#1): CREATE mRNA (3 µg per well) and sgRNAs (0.121 µg each sgRNA per well) were transfected using Lipofectamine MessengerMAX (Invitrogen).
- Two-step, mRNA first Protocol (#2): CREATE mRNA (3 µg per well) was transfected using Lipofectamine MessengerMAX. Transfected cells were incubated at 37 °C with 5% $CO_2$ for 4 h. After refreshing the media, cells were then transfected with sgRNAs (0.121 µg each sgRNA per well) using Lipofectamine RNAiMax (Invitrogen).
- Two-step, sgRNA first Protocol (#3): sgRNAs were transfected using Lipofectamine RNAiMax. Transfected cells were incubated at 37 °C with 5% $CO_2$ for 4 h. After refreshing the media, cells were then transfected with CREATE mRNA (3 µg per well) using Lipofectamine MessengerMAX.

## Electroporation

Huh7 or 293T cells were harvested by TrypLE Select and resuspended in MaxCyte buffer at a density of $6.25 \times 10^7$ cells/mL. For MaxCyte electroporation, 15 µg of CREATE mRNA, 7.5 µg of nCas9 mRNA, and 2.5 µg/each sgRNAs were gently mixed with $1.25 \times 10^6$ cells in a total volume of 27.5 µL MaxCyte buffer. 25 µL of cell-RNA mixture was transferred into an OC-25x3 cuvette. Electroporation was carried out on the MaxCyte ATx following manufacture's instruction. Immediately after electroporation, the OC-25x3 cuvettes with cells were incubated at 37 °C for 15 min. Cells were then added to a 6-well plate with DMEM with 5% FBS for the first 3 days after electroporation. The medium was removed and switched back to DMEM with 10% FBS after day 3. Cells were harvested on day 3 and day 8 post-transfection for FACS analysis. Electroporation of activated human T cells were performed using

the MaxCyte ATx with OC-25x3 cuvettes by following the same procedure as described above with a T cell specific electroporation program. Following electroporation, T cells were cultured in TexMACS media with 5% heat-inactivated human AB serum, 10 ng/mL IL-2, 5 ng/mL IL-7, and 5 ng/mL IL-15 (ThermoFisher).

## Flow cytometry analysis and cell sorting

Flow cytometry analysis was performed on day 3 and day 8 after transfection. Transfected cells were collected after PBS washing and TrypLE Select digestion and resuspended in PBS with 1% FBS and 2 mM EDTA for flow cytometry analysis (CYTEK). Data were analyzed by FlowJo software.

For experiments that require enriched GFP-positive cells, cells were collected three to five days after transfection, and sorted by flow cytometry (BD FACSMelody) to obtain the GFP-positive cells. Sorted cells were cultured in DMEM with 20% FBS for the initial 7 days to facilitate cell recovery.

## Genomic DNA extraction

One million collected cells were centrifuged at $5000 \times g$ for 5 min at 4 °C and the pellet was resuspended in 100 µL cold PBS. Genomic DNA was extracted using Monarch Genomic DNA Purification Kit (NEB) according to manufacturer's instructions and eluted in 20 µL water.

## Target hybridization sequencing

Target hybridization sequencing was performed at Azenta Life Sciences (https://www.azenta.com/). Genomic DNA (gDNA) extracted was used to prepare Insertion Site Analysis (ISA) libraries using the Twist Target Enrichment procedure. gDNA was enzymatically fragmented, followed by end repair, dA-tailing and adapter ligation, and library amplification using indexed primers. The resulting pre-hybridization libraries underwent QC followed by hybridization-capture with baits corresponding to the transgene/insert, and a final library amplification using universal primers. The final library was sequenced by Illumina platform using $2 \times 150$ bp sequencing configuration in paired-end mode.

For data analysis, raw BCL files generated by the sequencer were converted to raw fastq files for each sample using bcl2fastq v.2.20. Fastq reads were then trimmed with adapter sequences. Homology analysis was performed for vector genome against human reference genome (build hg38) and high homology regions identified in reference human genome were hard masked. A combined reference genome consisting of vector genome and hard masked human reference was generated and reads were aligned to the combined genome using bwa v0.7.4. The bam file was subsequently processed with Samtools v0.1.16. Finally, integration sites were identified by taking into account pairs of reads that support insertions and alignment to both vector and human reference genomes. Identified potential insertion sites were clustered and annotated by Azenta proprietary ISA pipeline. Potential insertion sites with >5 confirmed reads per location were analyzed and plotted. Detailed analysis of identified insertions sites with >5 confirmed reads are shown in Appendix Table S1.

## Illumina short-read amplicon sequencing

Genomic DNA samples were PCR amplified with Q5 High-Fidelity 2X PCR Master Mix (NEB) based on the manufacturer's protocol. PCR amplicon primers designed specifically to amplify integration locus-CREATE junctions are listed in Appendix Table S3. Amplicons were analyzed by 1% agarose gel electrophoresis and purified according to the sizes using the DNA Gel Extraction Kit (NEB). Amplicons of specific insertion junctions were prepared for sequencing using the Illumina platform. Alignment of amplicon reads to a reference sequence was performed using CRISPResso2. Reads with a mean quality score <30 were discarded.

## Nanopore long-read amplicon sequencing

Genomic DNA samples were PCR amplified with Q5 High-Fidelity 2X PCR Master Mix (NEB) based on the manufacturer's protocol. PCR amplicon primers designed specifically to amplify the entire payload transgene are listed in Appendix Table S3. Nanopore long-read amplicon sequencing was performed at Plasmidsaurus (https://plasmidsaurus.com). Amplicons were gel purified and library was prepared with Native Barcoding Kit 96 V14 (Nanopore, SQK-NBD114.96) and sequenced with R104.1. flow cell on PromethION P24 sequencer.

For analysis of sequencing results, expected post-integration reference genomic sequences for each target loci were created. Nanopore reads were aligned to the reference genome using minimap2 (v2.28-r1209) with the following parameters: -A 29 (match score), -B 17 (mismatch penalty), -O 25 (gap opening penalty), -E 2 (gap extension penalty), and -t 7 (to utilize 7 threads for parallel processing). After alignment, bedtools intersect was used to identify aligned reads with transgene integration between the two PBS sites in the reference genome by applying the options -F 1.0 -wa. The retained reads were then processed using pysam to extract and analyze the CIGAR string information. Only the portions of the reads that aligned between the two PBS sites were considered, and the CIGAR string was parsed to count the occurrences of Match (M), Insertion (I), and Deletion (D) events. These counts were subsequently used to generate histograms depicting the distribution of these alignment features.

## Determination of CREATE editing activity at potential off-target sites

To evaluate the off-target activity of CREATE editing, the potential off-target sites of a single sgRNA were detected by targeted PCR amplification. Potentially top-ranking predicted off-target sites of spacers were selected using the CRISPR-Cas9 guide RNA design checker (Integrated DNA Technologies). PCR primers and predicted off-target insertion are listed in Appendix Table S4. The PCR reactions were performed using Q5 High-Fidelity 2X PCR Master Mix (NEB) with optimized Tm and the PCR products were detected by 1% agarose gel.

## Statistical analysis

Statistical analysis was performed using GraphPad Prism v10. Figure legends describe statistical methods and significance values.

No samples were excluded from the experiments performed. No blinding was done during the experiments.

## Data availability

NGS sequencing data from this publication have been submitted to NCBI BioProject with the Accession Number: PRJNA1189019.

The source data of this paper are collected in the following database record: biostudies:S-SCDT-10_1038-S44319-024-00364-7.

## Peer review information

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

## Acknowledgements

We thank R. Vale (HHMI Janelia) and S. Mukherjee (Columbia U) for critical reading of the manuscript and the entire Myeloid Therapeutics team for support. We thank Xiang Li (Luffing Future LLC) for performing Long-read NGS data analysis. This work is fully funded by Myeloid Therapeutics.

## Author contributions

**Yuxiao Wang**: Conceptualization; Resources; Formal analysis; Supervision; Investigation; Visualization; Methodology; Writing—original draft; Writing—review and editing. **Ruei-Zeng Lin**: Conceptualization; Data curation; Formal analysis; Investigation; Visualization; Methodology; Writing—original draft; Writing—review and editing. **Meghan Harris**: Data curation; Formal analysis; Investigation; Methodology; Writing—original draft. **Bianca Lavayen**: Investigation; Visualization; Methodology. **Neha Diwanji**: Formal analysis; Investigation; Visualization; Methodology. **Bruce McCreedy**: Conceptualization. **Robert Hofmeister**: Writing—original draft; Project administration; Writing—review and editing. **Daniel Getts**: Conceptualization; Supervision; Funding acquisition; Writing—original draft; Project administration; Writing—review and editing.

Source data underlying figure panels in this paper may have individual authorship assigned. Where available, figure panel/source data authorship is listed in the following database record: biostudies:S-SCDT-10_1038-S44319-024-00364-7.

## Disclosure and competing interests statement

YW, RL, MH, BL, ND, RH, and DG are current employees of Myeloid Therapeutics. BM was a past employee of Myeloid Therapeutics. All authors hold equity interest in Myeloid Therapeutics.

# Expanded View Figures

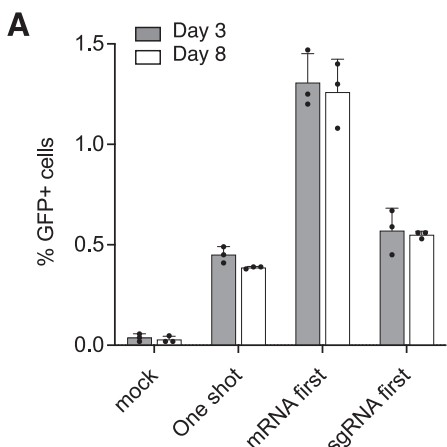

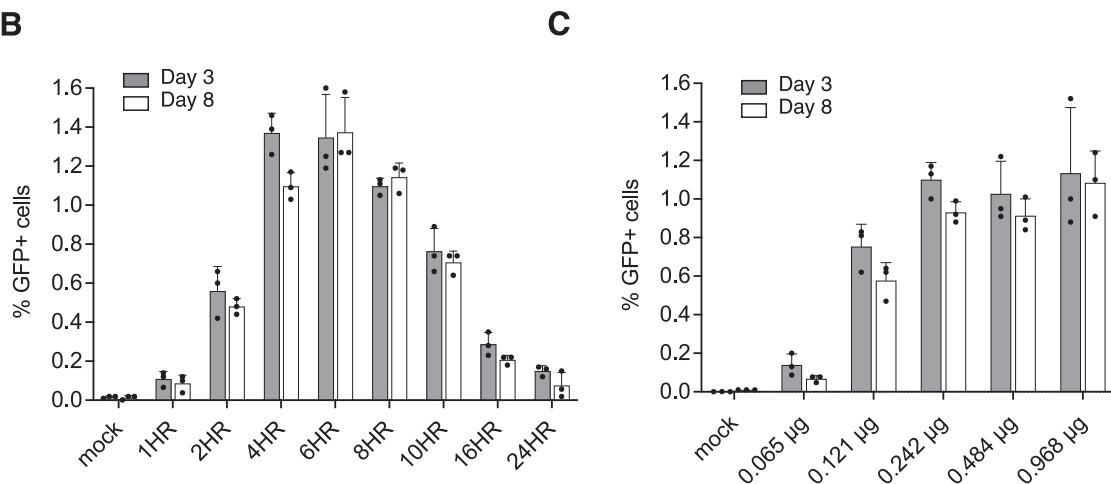

**Figure EV1.  Optimization of transfection protocol to improve CREATE editing efficiency.**

(**A**) Optimization of transfection protocol improved editing efficiency. Protocol 1 (one shot), Protocol 2 (mRNA first) and Protocol 3 (sgRNA first). (**B**) Optimization of the incubation time between mRNA and sgRNA transfections. (**C**) Optimization of the total amount of sgRNAs transfected. Data are mean ± SD ($n = 3$ biological replicates).

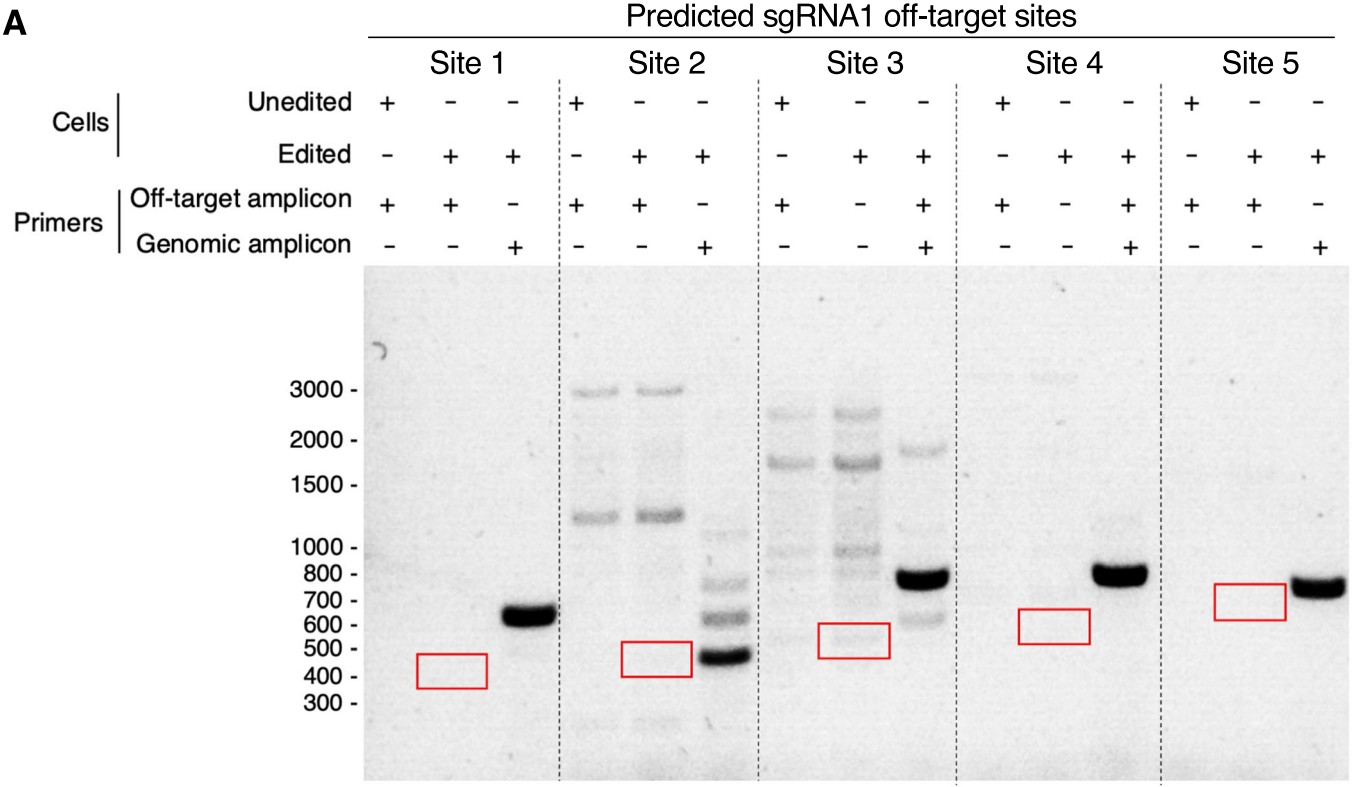

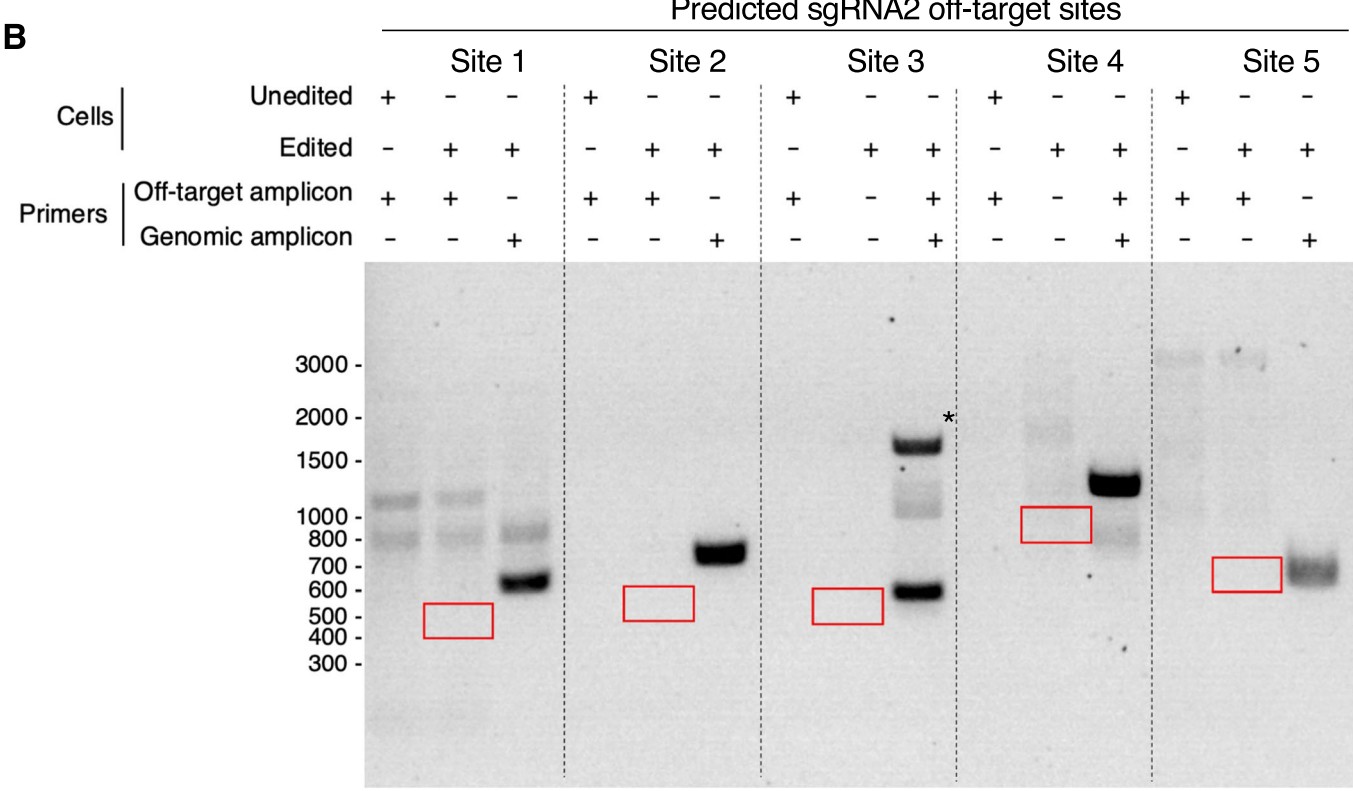

◄ **Figure EV2.  PCR detection of potential off-target editing in AAVS1 loci edited cells.**

Top 5 predicted off-target sites for sgRNA1 and sgRNA2 were examined. Red box indicate the size of the expected PCR product if off-target integration occurred. * indicate non-specific band amplicon.

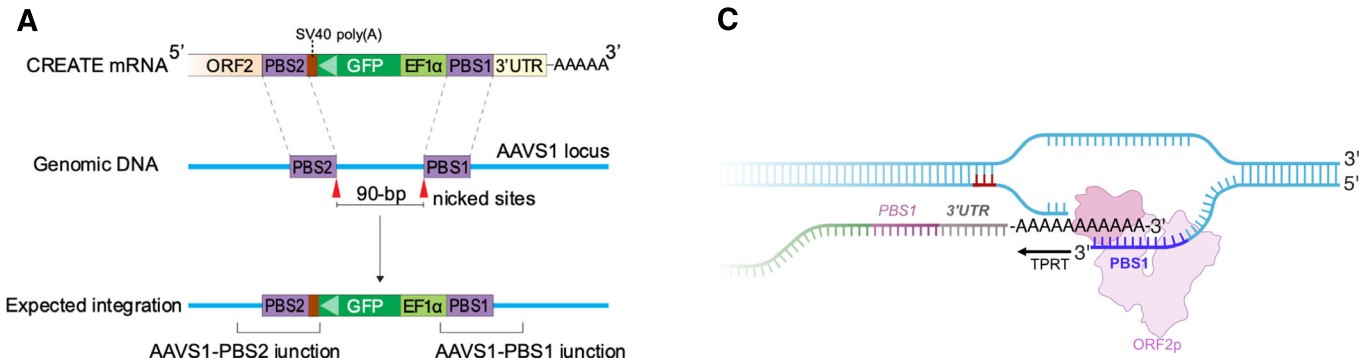

## B

**Top 5 indel type at AAVS1-PBS1 junction (17 bp PBS, Exp 1)**

| indel type | observation |
|---|---|
| EF1α PBS1 3'UTR AAVS1 PBS1<br>67-bp addition | Addition of 3'UTR, 35-bp AAVS1 genomic sequence, and PBS1 duplication |
| EF1α PBS1 3'UTR polyA PBS1<br>54-bp addition | Addition of 3'UTR, 11-bp polyA, and PBS1 duplication |
| EF1α PBS1 3'UTR<br>34-bp addition | Addition of 3'UTR and 15-bp unmapped sequence |
| EF1α PBS1 3'UTR polyA<br>61-bp addition | Addition of 3'UTR, 11-bp polyA, and 28-bp unmapped sequence |
| EF1α PBS1 3'UTR PBS1<br>29-bp addition | Addition of 3'UTR and PBS1 (truncated) duplication |

**Top 5 indel type at AAVS1-PBS1 junction (17 bp PBS, Exp 2)**

| indel type | observation |
|---|---|
| EF1α<br>74-bp deletion | Deletion of entire PBS1 and 50-bp of EF1a region |
| EF1α PBS1<br>38-bp deletion | 8-bp deletion of PBS1 and 30-bp in EF1a region |
| EF1α PBS1 3'UTR AAVS1 PBS1<br>46-bp addition | Addition of 3'UTR, 15-bp AAVS1 genomic sequence, and PBS1 duplication |
| EF1α PBS1 3'UTR<br>40-bp addition | Addition of 3'UTR and 22-bp unmapped sequence |
| EF1α PBS1 3'UTR polyA<br>48-bp addition  68-bp deletion | Addition of 3'UTR and 25-bp polyA, and 68-bp deletion in AAVS1 locus |

**Figure EV3. Analysis of the most frequently observed indels at PBS1 and PBS2 junctions.**

(A) A diagram showing expected transgene integration at AAVS1 locus. (B) Diagrams and descriptions of the most frequently observed indel patterns at PBS1 junctions in two independent experiments. (C) Potential mechanisms of alternative TPRT initiation that explain the observed indels.

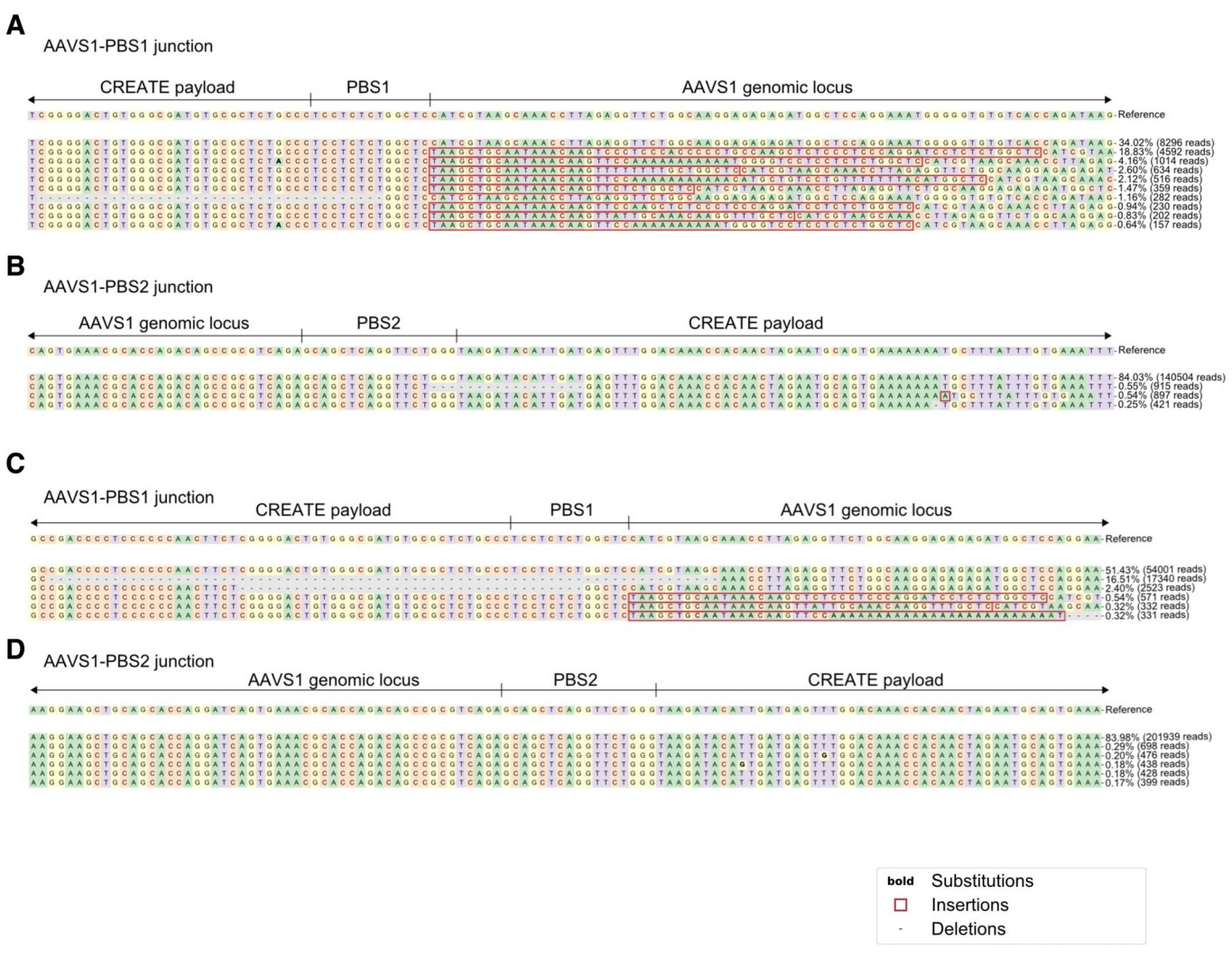

**Figure EV4. Allele plots of PBS1 and PBS2 junctions from NGS analysis of AAVS1 locus edited cells.**

(A) and (B) are from AAVS1 (17 bp PBS) Exp1 edited cells. (C) and (D) are from AAVS1 (17 bp PBS) Exp2 edited cells.

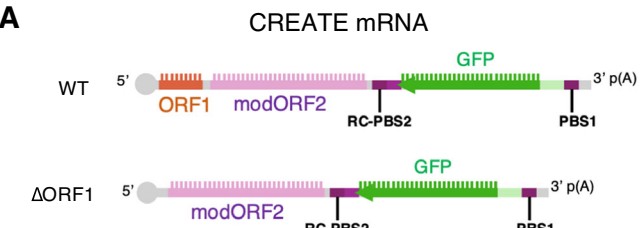

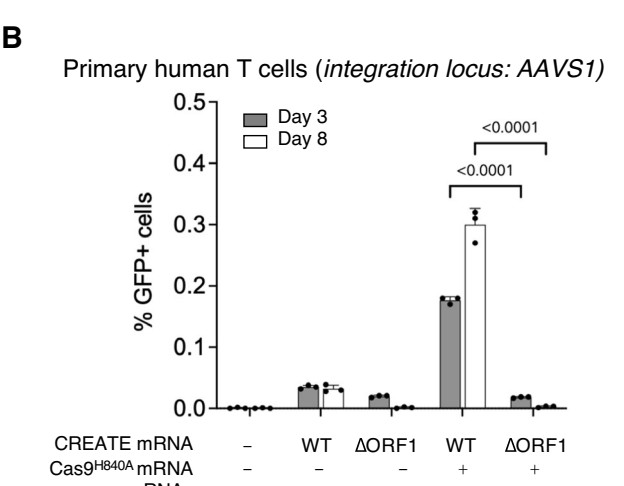

**Figure EV5. ORF1 is necessary for CREATE editing in T cells.**

(A) Diagrams of CREATE mRNA used in the experiment. (B) Deletion of ORF1 from CREATE mRNA (ΔORF1) abolished CREATE-mediated integration of GFP payload in primary T cells. Statistical analysis was performed using two-way ANOVA with Dunnett's multiple comparisons test comparing each sample against the sample with WT CREATE mRNA, nCas9$^{H840A}$ mRNA and sgRNAs. Data are mean ± SD ($n = 3$ biological replicates). Experiments were repeated twice. Exact $p$ values: <0.0001 (WT vs ΔORF1, Day 3), <0.0001 (WT vs ΔORF1, Day 8).

