## [Peer Review File · EMBO Reports]

CRISPR-Enabled Autonomous Transposable Element (CREATE) for RNA-based gene editing and delivery

Yuxiao Wang, Ruei-Zeng Lin, Meghan Harris, Bianca Lavayen, Neha Diwanji, Bruce McCreedy, Robert Hofmeister, and Daniel Getts

Corresponding authors: Yuxiao Wang (ywang@myeloidtx.com) , Daniel Getts (daniel@myeloidtx.com)

Review Timeline:

Submission Date:	12th Jul 24
Editorial Decision:	28th Aug 24
Revision Received:	28th Nov 24
Editorial Decision:	10th Dec 24
Revision Received:	14th Dec 24
Accepted:	19th Dec 24

Editor: Esther Schnapp

Transaction Report:

Dear Dr. Wang,

Thank you for the submission of your manuscript to EMBO reports. We have now received the full set of referee reports that is pasted below.

As you will see, the referees acknowledge that the findings are potentially interesting. However, they also have several suggestions for how the study should be improved, and I think all points are good and should be addressed. Please let me know in case you disagree, and we can discuss this further, also in a video chat, if you like.

I would thus like to invite you to revise your manuscript with the understanding that the referee concerns must be fully addressed and their suggestions taken on board. Please address all referee concerns in a complete point-by-point response. Acceptance of the manuscript will depend on a positive outcome of a second round of review. It is EMBO reports policy to allow a single round of major revision only and acceptance or rejection of the manuscript will therefore depend on the completeness of your responses included in the next, final version of the manuscript.

We realize that it is difficult to revise to a specific deadline. In the interest of protecting the conceptual advance provided by the work, we recommend a revision within 3 months (28th Nov 2024). Please discuss the revision progress ahead of this time with the editor if you require more time to complete the revisions.

- 1) A data availability section providing access to data deposited in public databases is missing. If you have not deposited any data, please add a sentence to the data availability section that explains that.
- 2) Your manuscript contains statistics and error bars based on $n=2$. Please use scatter blots in these cases. No statistics should be calculated if $n=2$.

5) a complete author checklist, which you can download from our author guidelines . Please insert information in the checklist that is also reflected in the manuscript. The completed author checklist will also be part of the RPF.

6) Please note that all corresponding authors are required to supply an ORCID ID for their name upon submission of a revised

manuscript (). Please find instructions on how to link your ORCID ID to your account in our manuscript tracking system in our Author guidelines

- the name of the statistical test used to generate error bars and P values,
- the number (n) of independent experiments (please specify technical or biological replicates) underlying each data point,
- the nature of the bars and error bars (s.d., s.e.m.),
- If the data are obtained from n {less than or equal to} 2, use scatter blots showing the individual data points.

12) All Materials and Methods need to be described in the main text using our 'Structured Methods' format, which is required for all research articles. According to this format, the Methods section includes a Reagents and Tools Table (listing key reagents, experimental models, software and relevant equipment and including their sources and relevant identifiers) followed by a Methods and Protocols section describing the methods using a step-by-step protocol format. The aim is to facilitate adoption of the methodologies across labs. More information on how to adhere to this format as well as a downloadable template (.docx) for the Reagents and Tools Table can be found in our author guidelines:

An example of a Method paper with Structured Methods can be found here: <https://www.embopress.org/doi/full/10.1038/s44320-024-00037-6#sec-4>.

As part of the EMBO publication's Transparent Editorial Process, EMBO reports publishes online a Review Process File (RPF) to accompany accepted manuscripts. This File will be published in conjunction with your paper and will include the referee

reports, your point-by-point response and all pertinent correspondence relating to the manuscript.

I look forward to seeing a revised form of your manuscript when it is ready. Please use this link to submit your revision:
<https://embor.msubmit.net/cgi-bin/main.plex>

Referee #1:

Wang and Lin et al. describe a new genome editing tool called CREATE that harnesses a repurposed L1 system plus a Cas9-nickase to enable all RNA mediated insertion of large payloads. A system that enables all RNA encoded large payload insertions remains one of the more exciting areas of therapeutic genome editing as the delivery of DNA donors needed for other gene insertion technologies such as PASTE, LSR recombinases and PASSIGE remains a challenge. As such CREATE should generate substantial interest in the genome editing community. CREATE seems to be highly specific for insertions at the targeted site but is not highly efficient with insertions in the low single digit range. The authors show CREATE works across 4 target sites and 3 cell types including primary T cells. The authors should be commended for doing a series of experiments with different mutations and truncations to the PBS, ORF1p, ORF2p or Cas9 nickase to demonstrate the proposed mechanism of CREATE insertions is supported. Overall CREATE is a very exciting technology and this should open the door for additional technology development or enable a variety of new research possibilities in biology or therapeutic approaches in medicine.

Minor points

- Please elaborate on any potential caveats based on the design of the hybridization probes used for NGS evaluation of CREATE specificity in Figure 1D. Will this approach only detect full length insertions or can this approach detect partial or abortive insertions of the cargo sequence?
- Please elaborate on the NGS amplicon sequencing in Figure 3. The authors comment on indel formation versus insertions at PBS1 and PBS2 but are all the insertions full length or do you see partial insertions where CREATE primes at PBS1 and initiates reverse transcription but the full cargo is not integrated to any reason?
- What locus was edited in Figure 4c?

Referee #2:

Review of EMBOR-2024-59975V1

CRISPR-Enabled Autonomous Transposable 1 Element (CREATE) for RNA-based gene editing and delivery

Authors: Yuxiao Wang^{1*†}, Rwei-Zeng Lin^{1†}, Meghan Harris¹, Bianca Lavayen¹, Neha Diwanji¹, Bruce McCreedy, Robert Hofmeister¹ and Daniel Getts^{1*}

In this manuscript, the authors have devised an RNA-based gene delivery tool that utilizes an endonuclease mutant version (D205A) of the human Long INterspersed Element-1 (LINE-1 or L1) retrotransposon together with a Cas9 (H840A) nickase and two sgRNAs to deliver an GFP reporter gene into the AAVS1 site in HEK293T cells and several other human cell lines.

A previous study by Tao and colleagues (Tao et al., *Nature Comm.*, 2022) showed that human L1 can mediate insertions into Cas9 mediated double strand breaks. They also showed that L1 can integrate into single strand breaks created by Cas9 nickase albeit very inefficiently. The authors of the current study appear to have increased the efficiency of L1 insertions into single strand nicks created by the Cas9 nickase, as well as increased specificity of L1 insertions into nicks using their CREATE system.

The CREATE RNA is essentially a modified L1 retrotransposition construct (Moran et al., *Cell*, 1996) that has been modified with a GFP reporter gene (Ostertag et al., *NAR*, 2000) in the 3' UTR (GFP lacks the intron used in retrotransposition assays) that is flanked by unique sequences called primer binding sites (PBS) that correspond to target sites within the AAVS1 locus that are 90 bp apart. The CREATE RNA is transfected with a nickase Cas9 mutant (H840A) that cleaves a single strand of genomic DNA

opposite the target guide RNA and two sgRNAs that target the genomic PBS site sequences within the AAVS1 locus. The innovation over the Tao et al., study is the use of the PBS sequences which create two nicks that are 90 bps apart in opposite strands of the DNA at the target AAVS2 locus and the inclusion of PBS sites in the CREATE RNA that flank the GFP reporter which are hypothesized to facilitate the integration of the GFP reporter gene between the genomic PBS sites in the AAVS1 locus.

Overall, the manuscript is logically presented, the methods are sufficiently detailed, and the data in general supports the conclusions. However, there are several issues that should be addressed, which are outlined below:

- 1) The authors of this study have utilized a Cas9 nickase (H840A) to create a pair of single strand DNA lesions on opposite strands of DNA at the AAVS1 locus. They also utilize an endonuclease mutant L1 ORF2 (D205A), which mediates insertion of the GFP reporter gene into the DNA lesions created by the Cas9 nickase. The authors conclude that the insertion is mediated via "natural" L1 TPRT. Although their data suggests that EN-deficient L1 ORF2 (D205A) is required to mediate the insertions after Cas9 nicking, it seems more likely that the insertion is mediated by the Endonuclease independent retrotransposition (ENi) pathway, which is an important aspect of L1 biology that is likely relevant to their study. Wild type or canonical TPRT involves the sequential nicking of the target DNA by the L1 EN followed by reverse transcription. In a 2002 study by Morrish and colleagues (Morrish et al., Nature Genetics, 2002), the authors characterized an endonuclease-independent L1 retrotransposition pathway (ENi) in which they showed that endonuclease deficient ORF2p (i.e., ORF2 D205A and H230A) L1 mutants could integrate into cells that contained DNA lesions. Later studies expanded on these findings demonstrating the ENi could occur at mammalian Telomeres (Morrish et al., Nature, 2007). Also relevant to the current study is a recent study (Tao et al., Nature Comm., 2022) that demonstrated that both wild type and EN mutant L1s can integrate into Cas9 lesions. It would be a welcome addition to the manuscript if the authors would reference these works and re-evaluate and discuss their findings in light of these studies.
- 2) The diagram in figure 1a is a bit too small and somewhat difficult to follow. It would be helpful if the authors could:
 - a. Enlarge the insertion model diagram
 - b. Indicate the location of the PBS target sites, PAM sites and cleavage sites of the Cas9 nickase
 - c. Label the 5' and 3' ends of all DNA/RNA strands including sgRNAs in all steps of diagram
- 3) The authors claim that the genomic sequence at the AAVS1 locus between the PBS sites is replaced (deleted) by integration of the GFP reporter gene. How do the authors know that the sequence between the PBS sites is deleted and how do they think the sequence might be deleted?
- 4) Related to the above question, the authors confirm the integration of the GFP gene into the AAVS1 locus on chromosome 19 using target hybridization and next gen sequencing. However, this data seems only to provide information regarding the general accuracy of the insertion into the AAVS1 locus on ch19 (Fig 1d). The authors need to fully characterize some of the insertions to confirm junction sites and verify whether the intervening sequence between the PBS sites has been deleted. Also, characterizing full insertions will help determine whether insertions occur via canonical TPRT or variations thereof (i.e., Endonuclease Independent L1 retrotransposition (ENi)).
- 5) In figure 3, the authors should provide some examples of indels and fully characterize at least some of the insertions.
- 6) LINE 89: "Natural L1 retrotransposition relies on nicking of the non-targeting DNA strand 90 by the EN domain of ORF2p". I am not sure what the authors mean by "natural" L1 retrotransposition. In canonical TPRT, the wild type L1 ORF2p EN cleaves a single strand of genomic DNA at a degenerate consensus sequence of 5'-TTTT/A-3' where the "/" indicates the EN cleavage site to expose a 3'-OH on the nicked strand of DNA. My understanding is that in the author's assay in which they utilize the ORF2p D205A mutant that lacks EN activity, the Cas9 nickase nicks a single strand of the DNA, thereby replacing the ORF2p EN activity.
- 7) LINES 54-55: "transcription (TPRT) process mediated by the RT domain of ORF2p that converts the L1 mRNA into cDNA." For the TPRT mechanism, please cite: Luan et al, 1993 (PMID: 7679954)
- 8) LINE 99: Delete or modify the statement: "safe non-replicative gene insertion" as the data does not support this assertion.
- 9) Abstract LINE 15: "LINE1" should be written as "LINE-1". LINE-1 is an abbreviation for Long INterspersed Element-1.
- 10) LINE 182: "These results are with a recent study investigating the poly-A length required for efficient TPRT by L1 ORF2" Can the authors please explain how the results obtained through modulating the PBS lengths relate to poly-A tail length.

Referee #3:

* General comments:

Targeted integration of a large gene is definitely important in the genome editing field. To date, although various tools including prime editing combined with recombinase (PASSIGE or PASTE), CRISPR-associated transposon (CAST), Bridge RNA technology, retrotransposon system have been developed, it is still unsatisfactory to conduct targeted gene insertion with high efficiency in human cells. In the present study, Wang et al. demonstrate a new type of targeted integration strategy using human LINE1 (L1) retrotransposon element combined with CRISPR-Cas9 system, named CRISPR-Enabled Autonomous Transposable Element (CREATE). It is very smart to inactivate endonuclease domain of ORF2 (modORF2) to reduce genome-wide off-target integration. Although average integration efficiencies were not high enough (approximately 1%) in human cells, it is valuable to open this method to the field. I would like to raise a few issues to strengthen this study.

* Specific comments:

1. As depicted in Figure 1a, the necessity of "ORF1p" is vague. The original role of ORF1p is an RNA-binding protein that interacts with L1 mRNA transcripts. However, in the present study, the L1 mRNA is provided outside the cells by the authors. I assume that ORF1p might not be necessary for the CREATE activity. It would be beneficial in terms of shorter length of L1 mRNA.
2. The CREATE requires double-nicking system by nCas9, instead of endonuclease of ORF2, which unfortunately results in unwanted indel mutations at the target site. Because it is not unavoidable, I should raise a fundamental question, "Is the CREATE a tool without causing a DNA double-strand break (DSB)?" Although the CREATE does not generate DSBs directly, double-nicking is typically accompanied with DSB.
3. In line 12, "double-stand breaks" -> "double-strand breaks"

Response to review of manuscript EMBOR-2024-59975V1

We sincerely thank all referees for their thorough review and insightful comments. Below is a point-by-point response. Blue colored text are the original comments.

Referee #1:

Wang and Lin et al. describe a new genome editing tool called CREATE that harnesses a repurposed L1 system plus a Cas9-nickase to enable all RNA mediated insertion of large payloads. A system that enables all RNA encoded large payload insertions remains one of the more exciting areas of therapeutic genome editing as the delivery of DNA donors needed for other gene insertion technologies such as PASTE, LSR recombinases and PASSIGE remains a challenge. As such CREATE should generate substantial interest in the genome editing community. CREATE seems to be highly specific for insertions at the targeted site but is not highly efficient with insertions in the low single digit range. The authors show CREATE works across 4 target sites and 3 cell types including primary T cells. The authors should be commended for doing a series of experiments with different mutations and truncations to the PBS, ORF1p, ORF2p or Cas9 nickase to demonstrate the proposed mechanism of CREATE insertions is supported. Overall CREATE is a very exciting technology and this should open the door for additional technology development or enable a variety of new research possibilities in biology or therapeutic approaches in medicine.

Minor points

- Please elaborate on any potential caveats based on the design of the hybridization probes used for NGS evaluation of CREATE specificity in Figure 1D. Will this approach only detect full length insertions or can this approach detect partial or abortive insertions of the cargo sequence?

Response: we thank the reviewer for this important question. The hybridization probes used for NGS were 120 nt probes tiled across the entire length of the payload gene. During library preparation, genomic DNA was enzymatically fragmented and captured by the probes, followed by Illumina short-read sequencing (150 bp x 2). This approach has limitations in distinguishing full-length vs partial insertions. While it can identify the presence of insertions at specific genomic loci, the short read lengths and variable probe capture efficiency prevent accurate quantification of full-length vs partial integrations.

To address this limitation, we have now performed Nanopore long-read sequencing of the amplicon covering the entire length of the integrated payload. This allows us to sequence full-length insertions and characterize partial integrations. We have included this new analysis in Fig. 4E and Appendix Fig. S3 and S4. Our results show that over 70% of the confirmed integration alignment showed full-length or near full-length transgene (< 20 bp total indel length).

- Please elaborate on the NGS amplicon sequencing in Figure 3. The authors comment on indel formation versus insertions at PBS1 and PBS2 but are all the insertions full length or do you see partial insertions where CREATE primes at PBS1 and initiates reverse transcription but the full cargo is not integrated to any reason?

Response: see above. We have performed Nanopore long read sequencing which addresses this question.

- What locus was edited in Figure 4c?

Response: we have clarified in Figure 4c that AAVS1 locus was targeted for integration of payload.

Referee #2:

In this manuscript, the authors have devised an RNA-based gene delivery tool that utilizes an endonuclease mutant version (D205A) of the human Long Interspersed Element-1 (LINE-1 or L1) retrotransposon together with a Cas9 (H840A) nickase and two sgRNAs to deliver an GFP reporter gene into the AAVS1 site in HEK293T cells and several other human cell lines.

A previous study by Tao and colleagues (Tao et al., Nature Comm., 2022) showed that human L1 can mediate insertions into Cas9 mediated double strand breaks. They also showed that L1 can integrate into single strand breaks created by Cas9 nickase albeit very inefficiently. The authors of the current study appear to have increased the efficiency of L1 insertions into single strand nicks created by the Cas9 nickase, as well as increased specificity of L1 insertions into nicks using their CREATE system.

The CREATE RNA is essentially a modified L1 retrotransposition construct (Moran et al., Cell, 1996) that has been modified with a GFP reporter gene (Ostertag et al., NAR, 2000) in the 3' UTR (GFP lacks the intron used in retrotransposition assays) that is flanked by unique sequences called primer binding sites (PBS) that correspond to target sites within the AAVS1 locus that are 90 bp

apart. The CREATE RNA is transfected with a nickase Cas9 mutant (H840A) that cleaves a single strand of genomic DNA opposite the target guide RNA and two sgRNAs that target the genomic PBS site sequences within the AAVS1 locus. The innovation over the Tao et al., study is the use of the PBS sequences which create two nicks that are 90 bps apart in opposite strands of the DNA at the target AAVS2 locus and the inclusion of PBS sites in the CREATE RNA that flank the GFP reporter which are hypothesized to facilitate the integration of the GFP reporter gene between the genomic PBS sites in the AAVS1 locus.

Overall, the manuscript is logically presented, the methods are sufficiently detailed, and the data in general supports the conclusions. However, there are several issues that should be addressed, which are outlined below:

1) The authors of this study have utilized a Cas9 nickase (H840A) to create a pair of single strand DNA lesions on opposite strands of DNA at the AAVS1 locus. They also utilize an endonuclease mutant L1 ORF2 (D205A), which mediates insertion of the GFP reporter gene into the DNA lesions created by the Cas9 nickase. The authors conclude that the insertion is mediated via "natural" L1 TPRT. Although their data suggests that EN-deficient L1 ORF2 (D205A) is required to mediate the insertions after Cas9 nicking, it seems more likely that the insertion is mediated by the Endonuclease independent retrotransposition (ENi) pathway, which is an important aspect of L1 biology that is likely relevant to their study. Wild type or canonical TPRT involves the sequential nicking of the target DNA by the L1 EN followed by reverse transcription. In a 2002 study by Morrish and colleagues (Morrish et al., *Nature Genetics*, 2002), the authors characterized an endonuclease-independent L1 retrotransposition pathway (ENi) in which they showed that endonuclease deficient ORF2p (i.e., ORF2 D205A and H230A) L1 mutants could integrate into cells that contained DNA lesions. Later studies expanded on these findings demonstrating the ENi could occur at mammalian Telomeres (Morrish et al., *Nature*, 2007). Also relevant to the current study is a recent study (Tao et al., *Nature Comm.*, 2022) that demonstrated that both wild type and EN mutant L1s can integrate into Cas9 lesions. It would be a welcome addition to the manuscript if the authors would reference these works and re-evaluate and discuss their findings in light of these studies.

Response: we thank the reviewer for highlighting the work on ENi retrotransposition and its potential relevance to our study. We agree that our initial use of the term "natural" L1 TPRT was

imprecise and potentially misleading. We have revised our description to more accurately reflect the CREATE mechanism, and provided a detailed step-by-step diagram showing the proposed mechanism (Fig. 1B). We have added a new section in the discussion that contextualizes CREATE within the broader landscape of L1 biology, including ENi and integration at Cas9-induced breaks. We now cite and discuss the works by Morrish et al. (2002, 2007) and Tao et al. (2022).

2) The diagram in figure 1a is a bit too small and somewhat difficult to follow. It would be helpful if the authors could:

a. Enlarge the insertion model diagram

b. Indicate the location of the PBS target sites, PAM sites and cleavage sites of the Cas9 nickase

c. Label the 5' and 3' ends of all DNA/RNA strands including sgRNAs in all steps of diagram

Response: we have made the requested changes and updated Fig. 1B.

3) The authors claim that the genomic sequence at the AAVS1 locus between the PBS sites is replaced (deleted) by integration of the GFP reporter gene. How do the authors know that the sequence between the PBS sites is deleted and how do they think the sequence might be deleted?

Response: we have added a detailed integration mechanism depiction in Fig. 1B and also clarified in the main text. Briefly, ORF2p synthesizes new DNA strands primed from PBS1 and PBS2 that anneal together and contains the payload; excision of the original, unedited DNA strands and ligation of the nicks would result in the replacement of the 90 bp sequences between the nicks with the payload (Fig. 1B)

4) Related to the above question, the authors confirm the integration of the GFP gene into the AAVS1 locus on chromosome 19 using target hybridization and next gen sequencing. However, this data seems only to provide information regarding the general accuracy of the insertion into the AAVS1 locus on ch19 (Fig 1d). The authors need to fully characterize some of the insertions to confirm junction sites and verify whether the intervening sequence between the PBS sites has been deleted. Also, characterizing full insertions will help determine whether insertions occur via canonical TPRT or variations thereof (i.e., Endonuclease Independent L1 retrotransposition (ENi)).

Response: to characterize the full insertions and analyze the junction site structure, we have performed further NGS analysis and Nanopore long-read sequencing. The results have been added

as Fig. 4E, EV3, EV4 and Appendix Figure S1-S4. We also discussed the potential insertion mechanism in light of this analysis.

5) In figure 3, the authors should provide some examples of indels and fully characterize at least some of the insertions.

Response: we have characterized the dominant indel patterns observed from amplicon sequencing. The analysis is added to the manuscript (Fig. EV3, EV4 and Appendix Fig. S1-S2)

6) LINE 89: "Natural L1 retrotransposition relies on nicking of the non-targeting DNA strand 90 by the EN domain of ORF2p". I am not sure what the authors mean by "natural" L1 retrotransposition. In canonical TPRT, the wild type L1 ORF2p EN cleaves a single strand of genomic DNA at a degenerate consensus sequence of 5'-TTTT/A-3' where the "/" indicates the EN cleavage site to expose a 3'-OH on the nicked strand of DNA. My understanding is that in the author's assay in which they utilize the ORF2p D205A mutant that lacks EN activity, the Cas9 nickase nicks a single strand of the DNA, thereby replacing the ORF2p EN activity.

Response: we thank the reviewer for the clarification. Indeed as the reviewer described, the CREATE relies on Cas9 to nick the non-target strand to release a 3' single-strand DNA flap, which then serve as the primer for TPRT by hybridizing with the PBS1 site on CREATE mRNA, reminiscent of the mechanism of canonical TPRT. We have modified the text to clarify this point and provided a detailed step-by-step diagram showing the proposed mechanism (Fig. 1B).

7) LINES 54-55: "transcription (TPRT) process mediated by the RT domain of ORF2p that converts the L1 mRNA into cDNA." For the TPRT mechanism, please cite: Luan et al, 1993 (PMID: 7679954)

Response: this citation is added.

8) LINE 99: Delete or modify the statement: "safe non-replicative gene insertion" as the data does not support this assertion.

Response: we have modified the sentence to remove this statement. The sentence now reads "The outcome of the editing cycle is the successful integration of the payload cassette between the PBS sites without insertion of retrotransposition competent L1 sequences."

9) Abstract LINE 15: "LINE1" should be written as "LINE-1". LINE-1 is an abbreviation for Long INterspersed Element-1.

Response: we have made the requested changes

10) LINE 182: "These results are with a recent study investigating the poly-A length required for efficient TPRT by L1 ORF2" Can the authors please explain how the results obtained through modulating the PBS lengths relate to poly-A tail length.

Response: we agree that our original statement was confusing and speculative. We have removed this sentence and instead focus on describing our empirical findings regarding optimal PBS length for CREATE efficiency. Below is a brief discussion regarding our original thoughts:

Thawani et al. investigated the optimal poly(A) length of mRNA template for ORF2p to initiate TPRT, and found that a poly(A) length of 20-50 nucleotides was optimal. This range closely aligns with our observed optimal PBS length of 20-30 nucleotides for CREATE efficiency (Thawani *et al*, 2023) . We speculated that the similarity in optimal length between the poly(A) tail for efficient TPRT by ORF2p and the PBS for CREATE is not coincidental. In CREATE, the PBS sequence may functionally mimic the role of the poly(A) tail by hybridizing with target DNA sequence and engage ORF2p for TPRT initiation.

Referee #3:

* General comments:

Targeted integration of a large gene is definitely important in the genome editing field. To date, although various tools including prime editing combined with recombinase (PASSIGE or PASTE), CRISPR-associated transposon (CAST), Bridge RNA technology, retrotransposon system have been developed, it is still unsatisfactory to conduct targeted gene insertion with high efficiency in human cells. In the present study, Wang et al. demonstrate a new type of targeted integration strategy using human LINE1 (L1) retrotransposon element combined with CRISPR-Cas9 system, named CRISPR-Enabled Autonomous Transposable Element (CREATE). It is very smart to inactivate endonuclease domain of ORF2 (modORF2) to reduce genome-wide off-target integration. Although average integration efficiencies were not high enough (approximately 1%) in human cells, it is valuable to open this method to the field. I would like to raise a few issues to strengthen this study.

Response: we thank the reviewer for the positive evaluation of the manuscript.

* Specific comments:

1. As depicted in Figure 1a, the necessity of "ORF1p" is vague. The original role of ORF1p is an RNA-binding protein that interacts with L1 mRNA transcripts. However, in the present study, the L1 mRNA is provided outside the cells by the authors. I assume that ORF1p might not be necessary for the CREATE activity. If successful without the ORF1p, it would be beneficial in terms of shorter length of L1 mRNA.

Response: we thank the reviewer for the suggestion. We tested a CREATE mRNA construct with ORF1p removed in primary T cells. Surprisingly, this construct failed to produce successful integration of the payload. This new data is now included in Fig. EV5. We think that ORF1p may serve several essential functions in the CREATE system:

a) RNP formation: ORF1p likely facilitates the assembly of CREATE ribonucleoprotein complexes, which may be necessary for efficient nuclear import and/or interaction with the target DNA (Freeman *et al*, 2019) .

b) Protection from cellular RNA degradation: ORF1p binding may shield the CREATE mRNA from cellular RNA decay mechanisms (Monot *et al*, 2013) .

2. The CREATE requires double-nicking system by nCas9, instead of endonuclease of ORF2, which unfortunately results in unwanted indel mutations at the target site. Because it is not unavoidable, I should raise a fundamental question, "Is the CREATE a tool without causing a DNA double-strand break (DSB)?" Although the CREATE does not generate DSBs directly, double-nicking is typically accompanied with DSB.

Response: we thank the reviewer for raising this important question. In Ran et al, 2013 the author used Cas9^{D10A} nickase with paired guide RNA to introduce DSB (Ran *et al*, 2013) . They systemically tested the off-set between the two sgRNAs and found that generally, an off-set of less than 100 bp are needed to observe DSB induced indels, and an off-set of > 150 bp will not cause spontaneous DSB formation. We propose that systemic testing of the off-set distance between the nicking sgRNA can identify the optimal design that maximize editing efficiency while minimizing DSB formation. We have added a discussion regarding DSB formation in the Discussion section.

3. In line 12, "double-stand breaks" -> "double-strand breaks"

Response: we corrected the mistake.

References:

- Freeman BT, Sokolowski M, Roy-Engel AM, Smither ME & Belancio VP (2019) Identification of charged amino acids required for nuclear localization of human L1 ORF1 protein. *Mobile DNA* 10: 20
- Monot C, Kuciak M, Viollet S, Mir AA, Gabus C, Darlix J-L & Cristofari G (2013) The Specificity and Flexibility of L1 Reverse Transcription Priming at Imperfect T-Tracts. *PLoS Genet* 9: e1003499
- Ran FA, Hsu PD, Lin C-Y, Gootenberg JS, Konermann S, Trevino AE, Scott DA, Inoue A, Matoba S, Zhang Y, *et al* (2013) Double Nicking by RNA-Guided CRISPR Cas9 for Enhanced Genome Editing Specificity. *Cell* 154: 1380–1389
- Thawani A, Ariza AJF, Nogales E & Collins K (2023) Template and target site recognition by human LINE-1 in retrotransposition. *Nature*: 1–3

Dear Dr. Wang,

Thank you for the submission of your revised manuscript. We have now received the enclosed reports from the referees and I am happy to say that all support its publication now. Only a few editorial requests will need to be addressed before we can proceed with the official acceptance of your manuscript:

- Your ms has 5 main figures but separate results and discussion sections. Please either add one more main figure or combine the results and discussion sections to publish your ms as a short report. The max character count for short reports is 30.000 including spaces but excluding Methods and References. Usually, combining results and discussions helps to eliminate some redundancy in the text so it becomes shorter.
- Please add up to 5 keywords to the ms file.
- Please correct the conflict of interest subheading to "Disclosure Statement and Competing Interests"
- Bruce McCreedy needs an affiliation number/sign superscript next to his name, please add.
- The author credits need to be removed from the ms file. All credits need to be entered during online ms submission.
- In the author checklist, please answer all questions on statistics and send us a new, completed list.
- Please also acknowledge all funders mentioned in the ms in our online submission system in the separate entries: Luffing Future LLC; Myeloid Therapeutics are currently missing.
- In the Appendix file, please add page numbers on the title page for all Appendix items.
- Please remove the Reagents & Tools table from the ms file and upload it as a separate file using the template from our GTA: <https://www.embopress.org/page/journal/14693178/authorguide#manuscriptpreparation>
Primer sequences can all be added to this table.
- Please upload all Source Data as one folder per figure.
- Materials and Methods should be called "Methods".
- Please provide the specific URL for the PRJNA1189019 dataset in the data availability section.
- Please provide the exact p values in the legends of figures 2B, 3A-C, 4A, 5B, C; EV5 B, as reasonable.
- Please write the abstract in present tense as per journal policy (demonstrated should be demonstrate).

EMBO press papers are accompanied online by A) a short (1-2 sentences) summary of the findings and their significance, B) 2-3 bullet points highlighting key results and C) a synopsis image that is exactly 550 pixels wide and 200-600 pixels high (the height is variable). The synopsis image should provide a sketch of the major findings, like a graphical abstract. Please note that text needs to be readable at the final size. Please send us this information along with the final manuscript.

Referee #1:

the authors have addressed all my concerns

Referee #2:

The authors have sufficiently addressed my comments.

Referee #3:

The authors have mostly answered the issues I raised in the earlier review. I recommend the publication of this revised version.

The authors addressed the remaining editorial issues.

Dr. Yuxiao Wang
Myeloid Therapeutics
300 Technology Square Suite 203
Cambridge, Massachusetts 02139
United States

Dear Dr. Wang,

I am very pleased to accept your manuscript for publication in the next available issue of EMBO reports. Thank you for your contribution to our journal.

Yours sincerely,
